# AttEXplore: Attribution for Explanation with model parameters eXploration

**Zhiyu Zhu**[1], **Huaming Chen**[1] [*], **Jiayu Zhang**[2], **Xinyi Wang**[3], **Zhibo Jin**[1],
**Jason Xue**[4] **& Flora D. Salim**[5]
University of Sydney[1], SuZhouYierqi[2], Universiti Malaya[3], CSIRO's Data61[4],
University of New South Wales[5]

## Abstract

Due to the real-world noise and human-added perturbations, attaining the trustworthiness of deep neural networks (DNNs) is a challenging task. Therefore, it becomes essential to offer explanations for the decisions made by these nonlinear and complex parameterized models. Attribution methods are promising for this goal, yet its performance can be further improved. In this paper, for the first time, we present that the decision boundary exploration approaches of attribution are consistent with the process for transferable adversarial attacks. Specifically, the transferable adversarial attacks craft general adversarial samples from the source model, which is consistent with the generation of adversarial samples that can cross multiple decision boundaries in attribution. Utilizing this consistency, we introduce a novel attribution method via model parameter exploration. Furthermore, inspired by the capability of frequency exploration to investigate the model parameters, we provide enhanced explainability for DNNs by manipulating the input features based on frequency information to explore the decision boundaries of different models. Large-scale experiments demonstrate that our **A**ttribution method for **E**xplanation with model parameter e**X**ploration (AttEXplore) outperforms other state-of-the-art interpretability methods. Moreover, by employing other transferable attack techniques, AttEXplore can explore potential variations in attribution outcomes. Our code is available at: https://github.com/LMBTough/ATTEXPLORE.

## 1 Introduction

Nowadays, DNNs have achieved state-of-the-art performance in various application scenarios such as medical diagnostics (Ribeiro et al., 2020), autonomous driving (Chen et al., 2021), and sentiment analysis (Tan et al., 2022). Given the usage in safety critical areas, the trustworthiness of such models plays a key role which may be affected by real-world noise and the human-added perturbations (Toreini et al., 2020; Jin et al., 2024; Zhu et al., 2024). Considering the intrinsic nonlinear and complex parameters nature, a trustworthy DNN model necessitates both high performance and interpretable decision making process (Adadi & Berrada, 2018; Maze et al., 2018; Small et al., 2023; Zhu et al., 2023a;b). Understanding the data propagation from model input to output is essential for Explainable Artificial Intelligence (XAI) (Sokol et al., 2023).

There are two different interpretation methods (Pan et al., 2021). Local approximation methods provide an explanation by approximating the local neighborhood behaviors of the target model at a particular point in the input space (Ribeiro et al., 2016; Shrikumar et al., 2017). Alternatively, gradient-based methods explain the target model via the gradients associated with the model inputs and provide the importance of the input features (Pan et al., 2021; Sundararajan et al., 2017). In this work, we focus on gradient-based methods, specifically attribution methods, which is to obtain pixel-level explanations determining the importance of each input feature for model decisions. Assuming that a small change to input features may alter the output, these features are considered an important factor aiding the sample in crossing the model's decision boundary, i.e., important features. Some

---

[*]Corresponding author: huaming.chen@sydney.edu.au

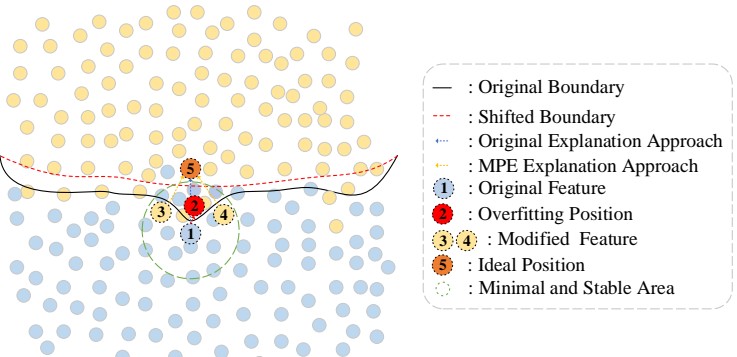

Figure 1: Decision boundary editing. Points ① to ② represent the traditional adversarial attack which may lead to overfitting. The region within the green dashed circle denotes the local instability of the decision boundary (where samples are relatively concentrated and cannot be reliably distinguished by the decision boundary). In order to obtain samples that cross the decision boundary more stably, points ① to ⑤ adjust the decision boundary to surmount the Shifted Boundary. Points ③& ④ to ⑤ simulate the effect of crossing the Shifted Boundary by generating modified samples.

recent methods are Integrated Gradients (IG) (Sundararajan et al., 2017), Boundary-based Integrated Gradient (BIG) (Wang et al., 2021b), Adversarial Gradient Integration (AGI) (Pan et al., 2021).

However, we need to consider the impact of inaccurate decision boundaries during model training since the training data is generally far from the decision boundaries. For the samples close to the decision boundary, they are likely to be OOD samples and are more sensitive. Avoiding such sensitive phenomena is crucial during the process of obtaining interpretive results. Besides, current gradient-based attribution methods may require either a baseline for integration (IG), or a specific linear integration path to quantify feature contributions (BIG). Even for AGI, the implemented adversarial attack is targeted, which may result in crossing multiple decision boundaries before reaching the target decision boundary, thus thwarting the interpretation particularly when there are similarities or overlaps between the decision boundaries of the target category and other categories. Therefore, we propose the *first* research question: **(i)** How to construct a **more general** decision boundary exploration approach to ensure that features can explore the current decision boundary?

As shown in Fig. 1, by modifying a portion of the features (yellow dots) so that they may cross the decision boundary, the most important features are found with minimal changes. We find that transferable attacks, which aim to obtain more transferable samples to perform black-box attacks, essentially consist of exploring model parameters to generate generic adversarial examples that can cross multiple decision boundaries. The decision boundaries obtained by transferable attacks are likely to be less overfitting, in other words, more accurate. This fits our idea of feature alterations to explore the current decision boundary in Fig. 1. Therefore, we propose to combine the decision boundary exploration method of transferable attacks with the attribution process, namely a novel model parameter exploration (MPE) based method, to obtain the needed feature changes.

To verify the integration of important features by the attribution algorithm, one way is to check whether the model makes correct decisions when only essential features are retained. However, this poses a challenge wherein a significant number of model parameters that should originally be activated remain inactive. It is worth noting that the inactive model parameters are primarily responsible for unimportant features, i.e., the decision boundary is shifted, but not too far. This necessitates that the attribution algorithm should exhibit strong stability and adaptability to different decision boundaries. Therefore, we propose the *second* research question: **(ii)** How to construct a **more robust** approach for minimal feature alterations via MPE to ensure the attribution performance?

Motivated by recent research demonstrating DNNs exhibit different sensitivities to different frequency domains for the human-added perturbations (Yin et al., 2019; Wang et al., 2020b; Guo et al., 2018), performing spectral transformations on inputs for frequency exploration provides new insights into model decision boundary exploration (Long et al., 2022). We find that the frequency information can significantly enhance the exploration of model parameters impacts on the decision

boundary. Moreover, exploring more model parameters leads to more precise attribution results. Therefore, we use frequency-based transferable attacks to generate minimally altered adversarial examples, and the results in the experimental section demonstrate the effectiveness of our approach. Notably, we are the first to introduce MPE to explore the decision boundaries of different models in a generalizable manner. Since different transferable attack methods explore the decision boundaries to varying degrees, our approach can be combined with other state-of-the-art transferable attack methods to discover potential variations in attribution performance. (See **Appendix A** for attribution results combining different state-of-the-art transferable attacks)

The main contributions of this paper are: **(1)** We uncover, for the first time, the decision boundary exploration approaches of attribution and transferable attacks are consistent. **(2)** We propose a novel attribution algorithm by performing Attribution for Explanation with Model Parameter Exploration based on transferable attacks, named AttEXplore. **(3)** We conduct extensive experiments to verify the effectiveness of our AttEXplore. **(4)** We release the code of AttEXplore publicly.

## 2 RELATED WORK

### 2.1 METHODS FOR INTERPRETING DNNS

**Local approximation methods** Local approximation methods typically ascertain an approximately interpretable surrogate model, thereby allowing the computation of gradient information and the derivation of attribution outcomes. For example, LIME (Ribeiro et al., 2016) amalgamates approximation techniques with weighted sampling methods to construct a local model for interpretable predictions. We note that LIME's interpretable behavior requires cluster segmentation of images, so it is not point-to-point. Shapley Additive Explanations (SHAP) (Lundberg & Lee, 2017) computes the contribution of each feature to the prediction outcome using Shapley values then ranks their importance. However, when applied to high-dimensional samples, SHAP typically incurs high computational complexity. DeepLIFT (Shrikumar et al., 2017) quantifies the significance of each input feature by elucidating the predictive influence on the deep learning model. However, its interpretation of nonlinear models is not necessarily accurate. In this paper, we prioritize gradient-based methods, as they are better suited for providing promising explanations on complex models.

**Gradient-based methods** Gradient information can be leveraged to visually represent the contribution values of image pixels, such as Grad-CAM (Selvaraju et al., 2017) and Score-CAM (Wang et al., 2020a). Saliency Map (SM) method (Simonyan et al., 2013) can produce interpretable results in non-CNN environments where CAM-based methods are not applicable, however, it is susceptible to gradient saturation, potentially yielding attribution results of zero.

To provide fine-grained pixel-level explanations, IG method (Sundararajan et al., 2017) rectifies the gradient deficiency observed in SM and introduces two axioms: *Sensitivity* and *Implementation Invariance*. By strategically selecting reference points as anchors along a linear integration path, IG integrates the continuous gradients to derive the attribution results. Following, BIG (Wang et al., 2021b) introduces a boundary search mechanism, resulting in more precise attribution outcomes. It resolves the concern of baseline selection process in IG. However, the integration path remains linear in BIG. AGI (Pan et al., 2021) further improves the performance by identifying the steepest non-linear ascending trajectory from the adversarial example $x'_i$ to $x$. Therefore, the attribution performance and stability hinge upon the quality of the adversarial samples.

Considering the integration path noise in IG, Guided Integrated Gradients (GIG) (Kapishnikov et al., 2021) obviates extraneous noisy pixel attributions by imposing constraints on the input and back-propagating gradients of the neurons, thus retaining only the pixel attributes pertinent to the predicted category. Nonetheless, it is limited to images, where the quality of input features significantly impacts the results, and the computational complexity is high. Other methods, such as Fast-IG (Hesse et al., 2021) and Expected Gradient (EG) (Erion et al., 2021), have similar concerns.

### 2.2 TRANSFERABLE ADVERSARIAL ATTACKS

The objective of transferable adversarial attacks is to craft general adversarial samples from the source model, capable of crossing decision boundaries across different models. Many algorithms have been proven to generate highly transferable adversarial samples. MI-FGSM (Dong et al., 2018)

and PGD (Madry et al., 2017) utilize advanced gradient calculations to improve the transferability of adversarial samples. Based on input transformation, SINI-FGSM (Lin et al., 2019), DI-FGSM (Xie et al., 2019), and TI-FGSM (Dong et al., 2019) adopt the image transformation methods on the input image to generate more transferable adversarial samples. As one of feature-level transferable attacks, NAA (Zhang et al., 2022) estimates the importance of intermediate layer neurons through neuron attribution, thereby solving the problem of inaccurate estimation of neuron importance by FDA (Ganeshan et al., 2019), FIA (Wang et al., 2021a) and other feature-level methods (Huang et al., 2019; Naseer et al., 2018).

By exploring potential model parameters with frequency information (Wang et al., 2020b; Guo et al., 2018; Yin et al., 2019), spectral transformation is implemented in (Long et al., 2022) for the input, effecting model augmentation to improve sample transferability. Thus, we consider harnessing the power of the frequency information to further explore different decision boundaries.

# 3 PRELIMINARIES

## 3.1 AXIOMS OF SENSITIVITY AND IMPLEMENTATION INVARIANCE

The beauty of attribution methods is from the axioms (Sundararajan et al., 2017). Since our method maintains a one-to-one correspondence between model inputs and outputs during attribution, it also satisfies these two axioms. Detailed proofs are provided in the **Appendix B**.

**Sensitivity** An attribution method adheres to the axiom of *Sensitivity* when, for any given input and baseline instances differing solely in one feature yet yielding distinct predictions, said divergent feature is allocated a non-zero attribution.

**Implementation Invariance** An attribution method conforming to the axiom of *Implementation Invariance* should guarantee that two neural network attributions, when applied to identical input and output values, exhibit consistency.

## 3.2 DEFINITION OF DECISION BOUNDARIES

The decision boundary refers to a hyperplane, curve, or boundary that separates data points of different classes or sets in the input data space (Shalev-Shwartz & Ben-David, 2014). The position, shape, and characteristics of the decision boundary depend on the model's structure and its parameters. Constructing a robust method for exploring and visualizing the decision boundaries of different DNN models is pivotal for understanding the decision-making process.

## 3.3 IG AND AGI METHODS

Formally, in order to explicate the DNN model denoted as $f(\cdot)$, we define the input feature $x \in \mathbb{R}^n$, where $n$ is the dimension of the input feature, and the model output is represented as $Y = f(x)$. The primary objective of attribution lies in the determination of $A \in \mathbb{R}^n$, which is to elucidate the corresponding significance of each feature within $x$. According to Saliency Map (Simonyan et al., 2013), if a DNN model $f$ exhibits continuous differentiability, the input feature importance measure $A$ can be derived from the gradient information $\frac{\partial f}{\partial x}$. It is imperative to underscore that this process engenders a one-to-one correspondence. For example, denote the input feature importance of IG by $IG_j(x)$, then the formula of IG is expressed in Eq.1.

$$A_j = IG_j(x) = (x_j - x'_j) \times \int_{\alpha=0}^{1} \frac{\partial f(x' + \alpha \times (x - x'))}{\partial x_j} \, d\alpha \tag{1}$$

where $j = 1, ..., n$ denotes the $j$-th input feature, $\frac{\partial f(x' + \alpha \times (x - x'))}{\partial x_j}$ is the gradient of model $f$ w.r.t input feature $x_j$. Here $x'_j$ represents the reference input feature. If we denote the input feature importance of AGI by $AGI_j(x)$, then the formula is described in Eq.2.

$$A_j = AGI_j(x) = AGI_{j-1}(x) - \nabla_{x_j} f^i(x) \cdot \epsilon \cdot sign(\frac{\nabla_{x_j} f^i(x)}{|\nabla_x f^i(x)|}) \tag{2}$$

$\nabla_{x_j} f^i(x)$ means the gradient corresponding to false class label $i$. Step size is represented by $\epsilon$. Eq.2 integrates along the path until $argmax_l f^l(x) = i$. We can see that, the decision boundary

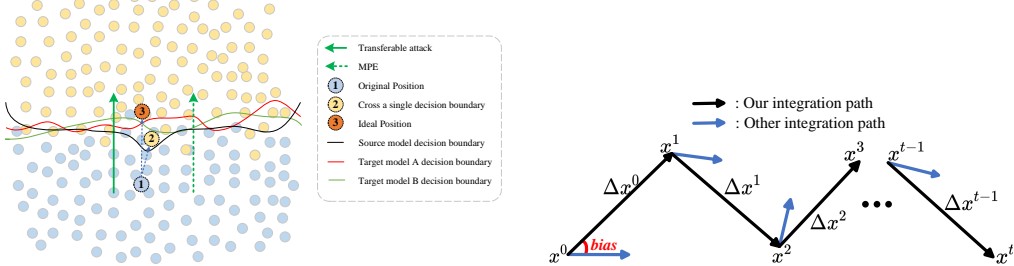

Figure 2: Decision boundaries          Figure 3: Our nonlinear integration path

exploration approach of IG is linear. For AGI, despite the non-linear decision boundary exploration approach without selecting specific reference points, it still needs to continuously cross the decision boundaries of other categories until the decision boundary category becomes $i$, which could potentially lead to overfitting issues and raise the concern of computation efficiency.

## 4 METHOD

### 4.1 EXPLORE DECISION BOUNDARIES VIA MODEL PARAMETER EXPLORATION (MPE)

**Feasibility of MPE** We discuss the relationship between model parameter exploration and attribution in this section. Since directly exploring the decision boundaries is difficult, we alternatively consider the model parameters to obtain the changes in model decision corresponding to the changes of a small number of parameters. This can significantly facilitate the attribution process. Assuming a model $y = L(x; w)$, where $y$ is the model output for input $x$ with the parameter $w$. Here we simplify the model to $y = w^T x$. If we consider a two-dimension scenario when $w = [1, 2]$, $x = [3, 4]$, $y = 11$. We have two methods to explore cases where the first parameter in $w^T$ is not activated. One method is to leave $w^T$ unchanged and the $x_0 = 0$, i.e., $x = [0, 4]$, at which point $y = 8$. Another method is to leave $x$ unchanged and $w_0^T = 0$, i.e., $w^T = [0, 2]$, at which point $y = 8$. We can see these two methods are equivalent, which means exploring $x$ is to some extent consistent with exploring $w^T$, i.e., $L(x; w)$ can be viewed as $L(w; x)$. Thus, model parameter exploration can be performed by modifying the input feature $x$ or adjusting the activation levels of parameters in $w$.

**MPE via transferable adversarial attacks** With the discussion of MPE, we understand that it is still infeasible to make extensive adjustments to the model's parameters, in particular, attribution algorithms aims to provide a rigorous explanation of the model's behaviour under current parameters. Moreover, due to the nature of attribution, in a scenario where a complete dataset is unavailable (Liu et al., 2014; Wang et al., 2020c; Retsinas et al., 2020), we cannot systematically adjust the parameters to explore the decision boundary in a controlled manner. Therefore, we resort to adjusting input features to explore the model's decision boundary, aiming to obtain more precise attribution results.

We firstly confirm that modifying samples to explore different decision boundaries aligns with the methods of transferable adversarial attacks. As illustrated in Fig. 2, the goal of transferable attacks is to generate samples with strong transferability on a local surrogate model to launch an attack on the target black-box model. Since different black-box models have different decision boundaries, developing a robust adversarial sample generation method to cross the decision boundaries is the core idea. Currently, input transformation-based transferable attacks represent the state-of-the-art (Lin et al., 2019; Xie et al., 2019; Dong et al., 2019), in which input samples are modified to generate general adversarial samples. This aligns with our idea of modifying features for model parameter exploration. Therefore, we propose to incorporate the transferable attack method in the attribution algorithm to enhance decision boundary exploration, as a solution to the ***first*** research question.

### 4.2 ATTRIBUTION FOR EXPLANATION WITH MODEL PARAMETER EXPLORATION (ATTEXPLORE)

**Novel nonlinear integration path** In AGI (Pan et al., 2021), the nonlinear integration path has been proven to be beneficial for attribution results. Specifically, nonlinear integration paths allow for

more accurate assignment of weights to features as well as capturing the nonlinear behaviour of the model in a more comprehensive way. In order to utilize model parameter exploration for attribution, we design a novel nonlinear integration path as in Fig. 3. We use Eq. 3 to mathematically explain our integration path, with detailed proofs in the **Appendix B**.

$$A = \int \triangle x^t \odot g(x^t) \mathrm{d}t \tag{3}$$

where $\triangle x^t$ represents the difference in the sample as it varies along the boundary in the decision direction. $g(x^t)$ denotes the gradient information that needs to be accumulated during the integration process. $y$ represents the original label. $\odot$ denotes hadamard product. There are two options for $g(x^t)$ in the integration process. One is the actual updated gradient obtained after MPE, corresponding to the black arrow in Fig. 3. The other is the gradient obtained by recomputing the current sample $x_f$, which corresponds to the blue arrow in Fig. 3. In BIG and AGI, it is expressed as $\frac{\partial L(x_{f_i}, y)}{\partial x_{f_i}}$. Taking AGI as an example, since it is a targeted attack, the model may cross multiple decision boundaries of other categories before reaching the decision boundary of a specific category. This results in slight biases in AGI's nonlinear integration path before the integration is completed, leading to unnecessary attacks and attributions (i.e., the angle of bias in Fig. 3). Therefore, in order to integrate the attribution results more smooth and robust in our nonlinear integration path, we use MPE from Eq. 3 to explore the decision boundary and update the gradient information of the model.

**Frequency-based input feature alterations method** Frequency domain information can effectively explore model parameters and generate highly transferable adversarial samples (Wang et al., 2020b; Guo et al., 2018; Yin et al., 2019), which can assist the attribution process. Inspired by SSA (Long et al., 2022), we propose a frequency-based input feature alterations method to generate input features that can effectively cross different decision boundaries, as detailed in Eq. 4- 6.

$$x_{f_i}^t = IDCT(DCT(x^t + N(0,1) \cdot \frac{\epsilon}{255}) * N(1, \sigma)) \tag{4}$$

$$\triangle x^t = \eta \cdot sign(\frac{1}{N} \sum_{i=1}^{N} \frac{\partial L(x_{f_i}^t, y)}{\partial x_{f_i}^t}) \tag{5}$$

$$g(x^t) = \frac{1}{N} \sum_{i=1}^{N} \frac{\partial L(x_{f_i}^t, y)}{\partial x_{f_i}^t} \tag{6}$$

From Eq. 4, to explore different frequency domains of the input feature $x$, we first use Discrete Cosine Transform (DCT) (Ahmed et al., 1974) to map the features into the frequency space. Then, we generate $N$ approximate features $x_{f_i}^t$ of $x^t$ by adding noise to the original features and applying random transformations in the frequency space. Here $\epsilon$ is the perturbation rate, and $i$ represents the number of frequency domain explorations. The inverse discrete cosine transformation (IDCT) serves as the reverse operation of DCT, allowing the image to be transformed back to the spatial domain. It is important to note that both DCT and IDCT operations are lossless, and they facilitate the ease of gradient calculations (Ahmed et al., 1974). From Eq.5, we randomly select $N$ approximate features and average the results to represent the difference in samples. $L$ represents the target model, $sign(\cdot)$ determines the direction of integration, and $\eta$ is the learning rate. Eq. 6 is the specific mathematical formula for gradient information calculation. We address the *second* research question by utilizing our novel nonlinear integration path and frequency-based input feature alterations method.

## 5 EXPERIMENTS

### 5.1 EXPERIMENTAL SETTINGS

**Dataset and Models** In this study, we employ ImageNet dataset (Deng et al., 2009). We conduct experiments on a selection of 1000 samples from ImageNet, guided by the principles outlined in NAA (Zhang et al., 2022), SSA (Long et al., 2022), and AGI (Pan et al., 2021). Furthermore, we employ three commonly utilized CNN models in the field of image classification: Inception-v3 (Szegedy et al., 2016), ResNet-50 (He et al., 2016), and VGG16 (Simonyan & Zisserman, 2014). Notably, we also employ the ViT-B/16 (Dosovitskiy et al., 2020) model to investigate the interpretability of our method on transformer-based visual models.

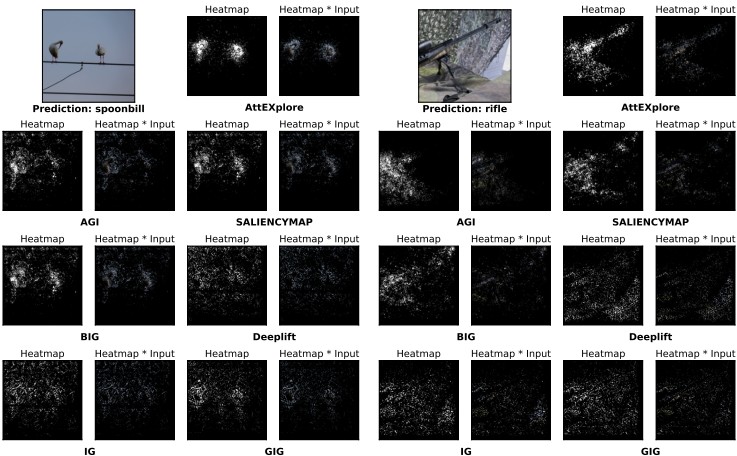

Figure 4: Visualization Results of our AttEXplore and Other Competitive Methods

**Baselines** We primarily compare with the state-of-the-art attribution algorithm, AGI (Pan et al., 2021). We also include eight other classical interpretability algorithms for comparative analysis, namely BIG (Wang et al., 2021b), DeepLIFT (Shrikumar et al., 2017), GIG (Kapishnikov et al., 2021), EG (Erion et al., 2021), Fast-IG (Hesse et al., 2021), IG (Sundararajan et al., 2017), SM (Simonyan et al., 2013), SG (Smilkov et al., 2017), and Grad-CAM (Selvaraju et al., 2017).

**Evaluated Metrics** We adhere to the evaluation metrics, specifically the Insertion&Deletion Scores, commonly employed by interpretability algorithms (Pan et al., 2021). Insertion Score quantifies the degree of change in model output when pixels are inserted into the input. A higher score signifies superior algorithmic interpretability. Conversely, Deletion Score measures the extent of model output change when pixels are removed from the input. A lower score indicates enhanced interpretability of the algorithm. It is noted that, for attribution algorithms, the importance of the Insertion Score outweighs that of the Deletion Score. This is due to the adversarial nature of neural networks, where Deletion Score may offer unreliable indications (Petsiuk et al., 2018). Hence, the Insertion Score serves as a more representative performance metric for attribution algorithms, while the Deletion Score can serve as an auxiliary metric to analyze attribution algorithms from multiple dimensions. Additionally, we employ the INFD score (Yeh et al., 2019) to demonstrate the faithfulness of our method to the underlying model. The lower the INFD score, the higher the faithfulness.

**Parameters** All experiments are conducted using an AMD Ryzen Threadripper PRO 5955WX 16-Core CPU, NVIDIA RTX6000 Ada GPU, and Ubuntu 22.04. Additionally, we apply the following general parameters setting: momentum set to 1.0, mask control parameter $\rho$ set to 0.5, number of approximate features $N$ set to 20, standard deviation of Gaussian noise ($\sigma$) set to 16, perturbation rate ($\epsilon$) set to 48/255, and total attack iterations (num_steps) set to 10. We notice a significant boosting is achieved without fine-tuning. Further parameter tuning may lead to much better performance.

## 5.2 EXPERIMENTAL RESULTS

Fig. 4 displays the visual results of our AttEXplore and other methods on Inception-v3 (See **Appendix G** for more visualization results). It is clear that the output heatmaps of AttEXplore are denser and clearer compared to methods like AGI, BIG, etc. This implies that the attribution results with high attribute values are more concentrated on the target object. Based on the results presented in Tab. 1, it is evident that our proposed method exhibits significant performance improvement over other classical interpretability algorithms. Particularly for the Insertion Score, it has surpassed both similar classical interpretability algorithms and AGI. Meanwhile, the Deletion Score remains consistently at a relatively low level, consistently outperforming AGI in comparative assessments. To provide specific instances, on the Inception-v3 model, relative to AGI, our method has an increase of 4.89% in Insertion Score and a decrease of 1.42% in Deletion Score. When compared with other algorithms on average, our method results in an improvement of 20.59% in the Insertion Score and

a reduction of 1.86% in the Deletion Score. On the ResNet-50 model, relative to AGI, our method shows an increase of 4.01% in the Insertion Score and a decrease of 1.72% in the Deletion Score. When compared with other algorithms on average, our method improves by 24.06% in Insertion Score and has a reduction of 2.42% in Deletion Score. Finally, on the VGG-16 model, relative to AGI, our method led to an increase of 6.01% in the Insertion Score and a decrease of 0.93% in the Deletion Score. When compared with other algorithms on average, our method results in an improvement of 19.05% in the Insertion Score and a reduction of 1.81% in the Deletion Score. Notably, different from traditional CNN models, Vision Transformers (ViTs) process images as sequences of patches, rendering them challenging to interpret. In the **Appendix C**, we conducted additional experiments on ViT-B/16 (Dosovitskiy et al., 2020), and the results further substantiate the superior performance achieved by our method. Also in the **Appendix D**, the INFD score tests demonstrate that our method exhibits the highest faithfulness.

Table 1: Insertion&Deletion score comparison of AttEXplore and other competitive baselines

| Method | Inception-v3 | | ResNet-50 | | VGG-16 | |
|---|---|---|---|---|---|---|
| | Insertion Score | Deletion Score | Insertion Score | Deletion Score | Insertion Score | Deletion Score |
| Grad-CAM | 0.4496 | 0.1084 | 0.2541 | 0.0942 | 0.3169 | 0.0841 |
| BIG | 0.3563 | 0.0379 | 0.2272 | 0.0415 | 0.1762 | 0.0303 |
| SaliencyMap | 0.3974 | 0.0422 | 0.256 | 0.048 | 0.2089 | 0.0323 |
| DeepLift | 0.216 | 0.0314 | 0.1246 | 0.0256 | 0.0827 | 0.0157 |
| GIG | 0.2584 | 0.0239 | 0.1308 | 0.0184 | 0.0859 | 0.0142 |
| EG | 0.2364 | 0.1656 | 0.256 | 0.2178 | 0.1959 | 0.1797 |
| Fast-IG | 0.146 | 0.0338 | 0.0889 | 0.0315 | 0.0623 | 0.0213 |
| IG | 0.2268 | 0.0284 | 0.1136 | 0.0247 | 0.0701 | 0.0173 |
| SG | 0.301 | 0.023 | 0.2357 | 0.0202 | 0.1423 | 0.015 |
| AGI | 0.4243 | 0.0439 | 0.3796 | 0.0465 | 0.2585 | 0.0319 |
| AttEXplore (ours) | 0.4732 | 0.0297 | 0.4197 | 0.0293 | 0.3186 | 0.0226 |

## 5.3 ANALYSIS OF TIME COMPLEXITY

We use FPS, the number of $f$rames processed by the algorithms $p$er $s$econd (FPS), to evaluate the algorithm processing speed (See **Appendix E** for the definition of FPS). All experiments are run in the same environment discussed in Section 5.1. We select five methods that closely match the performance of AttEXplore as our baselines. Other methods such as Saliency Map, DeepLIFT, Fast-IG, EG, and Grad-CAM demonstrate relatively poorer attribution accuracy compared to AttEXplore. Therefore, they are not considered for efficiency comparison. Table 2 demonstrates the superior computational efficiency of AttEXplore while also attaining enhanced attribution performance.

Table 2: FPS results for AttEXplore and state-of-the-art methods

| Method | BIG | AGI | IG | SG | GIG | AttEXplore |
|---|---|---|---|---|---|---|
| FPS | 3.3798 | 0.8818 | 19.7461 | 19.4942 | 2.2814 | 47.2805 |

## 5.4 ABLATION STUDY

Here we discuss the impact of three parameters, namely the approximate features number ($N$), the total attack iterations (num_steps), and the perturbation rate ($\epsilon$), on the performance of AttEXplore.

**Number of approximate features** ($N$) The total attack iterations are firstly fixed at 10, where the perturbation rate is 16. We change $N$ to values of 10, 20, 30, 40, 50, and 60, to assess the influence of this parameter on the performance of AttEXplore. In Table 3, with an increase in $N$, the performance of AttEXplore exhibits a gradual enhancement. Specifically, across three distinct models, namely Inception-v3, ResNet-50, and VGG-16, both insertion and deletion scores consistently increase as $N$ increases. It indicates that increasing the number of approximate features can effectively enhance the performance of AttEXplore. **Appendix F.1** contains results with additional values of $N$.

**Total attack iterations (num_steps)** We first keep $\epsilon$ at 16 and $N$ at 20. We then configure num_steps to be 5, 10, 15, 20, 25, and 30, to evaluate the influence on AttEXplore. Table 4 shows that, across three models of Inception-v3, ResNet-50, and VGG-16, there is a slight fluctuation in both insertion

Table 3: Insertion&Deletion score of AttEXplore with different values of $N$

| N | Inception-v3 | | ResNet-50 | | VGG-16 | |
|---|---|---|---|---|---|---|
| | Insertion Score | Deletion Score | Insertion Score | Deletion Score | Insertion Score | Deletion Score |
| 10 | 0.4603 | 0.0301 | 0.4004 | 0.0291 | 0.3074 | 0.0228 |
| 20 | 0.4644 | 0.0313 | 0.4022 | 0.0309 | 0.3096 | 0.0237 |
| 30 | 0.4649 | 0.0325 | 0.4033 | 0.0319 | 0.3090 | 0.0243 |
| 40 | 0.4659 | 0.0325 | 0.4045 | 0.0330 | 0.3108 | 0.0244 |
| 50 | 0.4665 | 0.0327 | 0.4032 | 0.0329 | 0.3118 | 0.0247 |
| 60 | 0.4679 | 0.0335 | 0.4037 | 0.0340 | 0.3107 | 0.0249 |

and deletion scores with an increase in num_steps. However, there is no evident trend indicating a significant impact of an augmented num_steps on the performance of AttEXplore. This suggests that, considering a set of $\epsilon$ and $N$, variations in num_steps exert a comparatively minor influence on the performance of AttEXplore. **Appendix F.2** contains results for different num_steps.

Table 4: Insertion&Deletion score of AttEXplore with different values of num_steps

| num_steps | Inception-v3 | | ResNet-50 | | VGG-16 | |
|---|---|---|---|---|---|---|
| | Insertion Score | Deletion Score | Insertion Score | Deletion Score | Insertion Score | Deletion Score |
| 5 | 0.4615 | 0.0307 | 0.3986 | 0.0287 | 0.3080 | 0.0224 |
| 10 | 0.4644 | 0.0313 | 0.4022 | 0.0309 | 0.3096 | 0.0237 |
| 15 | 0.4651 | 0.0324 | 0.4031 | 0.0322 | 0.3077 | 0.0244 |
| 20 | 0.4672 | 0.0329 | 0.4024 | 0.0331 | 0.3086 | 0.0244 |
| 25 | 0.4673 | 0.0332 | 0.4032 | 0.0336 | 0.3081 | 0.0248 |
| 30 | 0.4663 | 0.0339 | 0.4026 | 0.0339 | 0.3089 | 0.0252 |

**Perturbation rate ($\epsilon$)** We firstly fix $N$ at 20 and the num_steps at 10. We separately configured the perturbation rate ($\epsilon$) to be 8, 16, 24, 32, 40, and 48, to assess the influence on AttEXplore. Table 5 demonstrates that, across the three distinct models of Inception-v3, ResNet-50, and VGG-16, an increase in the perturbation rate is accompanied by a noticeable rise in the Insertion Score, while the Deletion Score exhibits a declining trend. This implies that in scenarios where the num_steps and $N$ remain relatively stable, a higher $\epsilon$ may be positively correlated with the performance of AttEXplore. The results with additional values of $\epsilon$ are included in **Appendix F.3**.

Table 5: Insertion&Deletion score of AttEXplore with different values of $\epsilon$

| $\epsilon$ | Inception-v3 | | ResNet-50 | | VGG-16 | |
|---|---|---|---|---|---|---|
| | Insertion Score | Deletion Score | Insertion Score | Deletion Score | Insertion Score | Deletion Score |
| 8 | 0.4637 | 0.0325 | 0.3962 | 0.0309 | 0.3065 | 0.0234 |
| 16 | 0.4644 | 0.0313 | 0.4022 | 0.0309 | 0.3096 | 0.0237 |
| 24 | 0.4659 | 0.0306 | 0.4071 | 0.0305 | 0.3121 | 0.0233 |
| 32 | 0.4675 | 0.0305 | 0.4109 | 0.0300 | 0.3142 | 0.0232 |
| 40 | 0.4714 | 0.0291 | 0.4157 | 0.0296 | 0.3161 | 0.0231 |
| 48 | 0.4732 | 0.0297 | 0.4197 | 0.0293 | 0.3186 | 0.0226 |

## 6 CONCLUSION

In conclusion, this paper introduces a novel method for Attribution for Explanation with model parameter eXploration (AttEXplore), which significantly advances the XAI results by providing enhanced interpretability for Deep Neural Networks (DNNs). Through the combination of model parameter exploration and frequency-based input feature alterations, AttEXplore outperforms state-of-the-art methods, demonstrating substantial improvements in both Insertion and Deletion Scores. By uncovering the relationship between attribution and transferable attack methods, we anticipate this work can contribute to a new standard for trustworthiness and explainability in deep neural networks. To achieve this, we also release the replication package of AttEXplore to facilitate improvements in future works. We hope this work will provide some insights to enhance the attribution method research community for a better XAI field.

ACKNOWLEDGMENT

Prof. Flora Salim acknowledges the support of the Australian Research Council (ARC) Centre of Excellence for Automated Decision-Making and Society (ADM+S) (CE200100005).

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

# A    ATTRIBUTION PERFORMANCE WHEN USING OTHER SOTA TRANSFERABLE ATTACKS FOR MODEL PARAMETER EXPLORATION

Table 6: Attribution performance with difference attack methods

| Method | Inception-v3 | | ResNet-50 | | VGG-16 | |
|---|---|---|---|---|---|---|
| | Insertion Score | Deletion Score | Insertion Score | Deletion Score | Insertion Score | Deletion Score |
| AttEXplore-NoAttack | 0.3959 | 0.0422 | 0.2584 | 0.0457 | 0.2121 | 0.0312 |
| AttEXplore | 0.4644 | 0.0313 | 0.4021 | 0.0308 | 0.3097 | 0.0237 |
| AttEXplore-PGD | 0.402 | 0.037 | 0.2901 | 0.0294 | 0.2258 | 0.0199 |
| AttEXplore-DI-FGSM | 0.4007 | 0.0416 | 0.2908 | 0.0427 | 0.2283 | 0.031 |
| AttEXplore-TI-FGSM | 0.3943 | 0.0344 | 0.3229 | 0.0348 | 0.2511 | 0.0268 |
| AttEXplore-MI-FGSM | 0.398 | 0.0414 | 0.2606 | 0.0447 | 0.2137 | 0.0304 |
| AttEXplore-SINI-FGSM | 0.4418 | 0.0314 | 0.3134 | 0.0292 | 0.2564 | 0.0227 |
| AttEXplore-NAA | 0.436 | 0.034 | 0.3058 | 0.0381 | 0.2559 | 0.0248 |

# B    DETAILED PROOFS OF TWO AXIOMS

Firstly, during the iterative process, the changes in the gradient along the integration path are captured by the original input information. Furthermore, it is not retroactive since feature values in previous iterations are unchanged in subsequent iterations. Therefore, the attribution result must be non-zero, which meets the definition of sensitivity. Here is the mathematical proof.

**Poof of Eq.3** :

We use the first-order Taylor approximation to expand the loss function and combine the information for the path from $\Delta x^0$ to $\Delta x^T$.

$$L\left(x^t\right) = L\left(x^{t-1}\right) + \frac{\partial L\left(x^{t-1}\right)}{\partial x^{t-1}}\left(x^t - x^{t-1}\right) + \varepsilon$$

$$\sum_{t=1}^{T} L\left(x^t\right) = \sum_{t=0}^{T-1} L\left(x^t\right) + \sum_{t=0}^{T-1} \frac{\partial L\left(x^t\right)}{\partial x^t}\left(x^{t+1} - x^t\right)$$

$$A = L\left(x^T\right) - L\left(x^0\right) = \sum_{t=0}^{T-1} \frac{\partial L\left(x^t\right)}{\partial x^t}\left(x^{t+1} - x^t\right) \qquad (7)$$

$$= \sum_{t=0}^{T-1} g(x^t) \odot \triangle x^t = \int \triangle x^t \odot g(x^t)\mathrm{d}t$$

Here $\epsilon$ is omitted due to the principle of higher-order Taylor expansions. And $\triangle x^t = x^{t+1} - x^t$, $g(x^t) = \frac{\partial L(x^t)}{\partial x^t}$.

Secondly, it is clear that the computational processes in AttEXplore follow the chain rule of gradients, which meets the definition of implementation invariance.

# C    ADDITIONAL EXPERIMENTS ON VIT-B/16

See Tab. 7.

# D    INFD SCORE TESTS

See Tab. 8.

# E    FPS DEFINITION

we use Frames Per Second (FPS) as an evaluation metric for our running efficiency. A higher FPS indicates a greater number of images generated per second, signifying a higher operational efficiency

Table 7: Attribution performance of AttEXplore and other competitive baselines on ViT-B/16

| Model | Method | Insertion Score | Deletion Score |
|---|---|---|---|
| ViT-B/16 | Saliency Map | 0.373 | 0.125 |
| ViT-B/16 | BIG | 0.422 | 0.093 |
| ViT-B/16 | GIG | 0.335 | 0.046 |
| ViT-B/16 | DeepLIFT | 0.296 | 0.063 |
| ViT-B/16 | EG | 0.361 | 0.329 |
| ViT-B/16 | Fast IG | 0.216 | 0.071 |
| ViT-B/16 | SG | 0.428 | 0.035 |
| ViT-B/16 | AGI | 0.425 | 0.069 |
| ViT-B/16 | IG | 0.346 | 0.051 |
| ViT-B/16 | AttEXplore (ours) | **0.470** | **0.062** |

Table 8: INFD Score

| Model | AGI | BIG | DeepLIFT | EG | Fast IG | GIG | IG | Saliency Map | SG | AttEXplore |
|---|---|---|---|---|---|---|---|---|---|---|
| Inception-v3 | 3.839 | 3.928 | 110.158 | 111.631 | 111.44 | 37.67 | 66.509 | 4.078 | 63.659 | 3.728 |
| ResNet-50 | 1.003 | 0.708 | 18.828 | 143.593 | 135.651 | 39.659 | 85.834 | 0.696 | 42.504 | 0.671 |
| VGG16 | 0.88 | 0.498 | 9.746 | 220.376 | 211.104 | 47.988 | 124.474 | 0.499 | 72.912 | 0.6 |

of the method.

$$FPS = \frac{Number\ of\ samples}{Running\ time\ of\ these\ samples} \tag{8}$$

# F  ADDITIONAL ABLATION STUDIES

## F.1  ABLATION STUDIES FOR THE NUMBER OF APPROXIMATE FEATURES

Table 9: Result for the number of approximate features ($N < 10$)

| N | Inception-v3 | | ResNet-50 | | VGG-16 | |
|---|---|---|---|---|---|---|
| | Insertion Score | Deletion Score | Insertion Score | Deletion Score | Insertion Score | Deletion Score |
| 1 | 0.4536 | 0.0282 | 0.3841 | 0.0262 | 0.2915 | 0.0190 |
| 2 | 0.4568 | 0.0284 | 0.3931 | 0.0275 | 0.2970 | 0.0196 |
| 3 | 0.4624 | 0.0298 | 0.3957 | 0.0274 | 0.3020 | 0.0210 |
| 4 | 0.4606 | 0.0292 | 0.3987 | 0.0286 | 0.3058 | 0.0214 |
| 5 | 0.4588 | 0.0300 | 0.3995 | 0.0282 | 0.3041 | 0.0223 |
| 6 | 0.4602 | 0.0288 | 0.3989 | 0.0289 | 0.3059 | 0.0221 |
| 7 | 0.4597 | 0.0301 | 0.3999 | 0.0288 | 0.3071 | 0.0224 |
| 8 | 0.4619 | 0.0301 | 0.4005 | 0.0289 | 0.3073 | 0.0229 |
| 9 | 0.4646 | 0.0299 | 0.3995 | 0.0291 | 0.3064 | 0.0225 |

We observe that when the perturbation rate is set to a larger value such as 48, the trend in model performance across different $N$ does not become more clear. This might be attributed to the fact that a larger perturbation rate represents a larger search space. Although the number of approximate samples increases, it doesn't mean that all these samples are necessarily effective for the attribution result.

Table 10: Result for the number of approximate features $N$ when the perturbation rate is 48

| Model | N | 1 | 2 | 3 | 4 | 5 | 6 | 7 | 8 | 9 | 10 | 20 | 30 | 40 | 50 | 60 |
|---|---|---|---|---|---|---|---|---|---|---|---|---|---|---|---|---|
| Inception-v3 | Insertion score | 0.459 | 0.461 | 0.465 | 0.465 | 0.464 | 0.467 | 0.468 | 0.47 | 0.472 | 0.466 | 0.471 | 0.473 | 0.473 | 0.474 | 0.474 |
| | Deletion score | 0.028 | 0.028 | 0.028 | 0.027 | 0.028 | 0.028 | 0.027 | 0.028 | 0.029 | 0.028 | 0.029 | 0.029 | 0.03 | 0.029 | 0.03 |
| ResNet-50 | Insertion score | 0.406 | 0.414 | 0.417 | 0.417 | 0.419 | 0.42 | 0.422 | 0.422 | 0.422 | 0.423 | 0.425 | 0.427 | 0.427 | 0.428 | 0.428 |
| | Deletion score | 0.027 | 0.028 | 0.028 | 0.029 | 0.03 | 0.029 | 0.03 | 0.03 | 0.03 | 0.03 | 0.031 | 0.031 | 0.032 | 0.032 | 0.032 |
| VGG16 | Insertion score | 0.298 | 0.304 | 0.306 | 0.308 | 0.308 | 0.308 | 0.31 | 0.31 | 0.311 | 0.311 | 0.312 | 0.313 | 0.313 | 0.313 | 0.315 |
| | Deletion score | 0.019 | 0.019 | 0.019 | 0.02 | 0.02 | 0.02 | 0.02 | 0.02 | 0.021 | 0.021 | 0.021 | 0.021 | 0.022 | 0.022 | 0.022 |

## F.2 ABLATION STUDIES FOR THE TOTAL ATTACK ITERATIONS

Table 11: Result for the total attack iterations num_steps

| | Inception-v3 | | ResNet-50 | | VGG-16 | |
|---|---|---|---|---|---|---|
| $num\_steps$ | Insertion Score | Deletion Score | Insertion Score | Deletion Score | Insertion Score | Deletion Score |
| 1 | 0.4236 | 0.0281 | 0.3469 | 0.0249 | 0.2645 | 0.0195 |
| 2 | 0.4488 | 0.0291 | 0.3835 | 0.0261 | 0.2929 | 0.0208 |
| 3 | 0.4557 | 0.0297 | 0.3936 | 0.0273 | 0.3029 | 0.0217 |
| 4 | 0.4607 | 0.0301 | 0.3966 | 0.0283 | 0.3061 | 0.0218 |

## F.3 ABLATION STUDIES FOR THE PERTURBATION RATE

Table 12: Result for the perturbation rate ($\epsilon < 8$)

| Model | $\epsilon$ | 1 | 2 | 3 | 4 | 5 | 6 | 7 |
|---|---|---|---|---|---|---|---|---|
| Inception-v3 | Insertion score | 0.459 | 0.459 | 0.459 | 0.461 | 0.46 | 0.461 | 0.46 |
| | Deletion score | 0.031 | 0.032 | 0.032 | 0.031 | 0.032 | 0.031 | 0.032 |
| ResNet-50 | Insertion score | 0.397 | 0.397 | 0.398 | 0.401 | 0.4 | 0.403 | 0.403 |
| | Deletion score | 0.031 | 0.031 | 0.031 | 0.032 | 0.032 | 0.032 | 0.032 |
| VGG16 | Insertion score | 0.298 | 0.299 | 0.3 | 0.3 | 0.3 | 0.302 | 0.303 |
| | Deletion score | 0.021 | 0.022 | 0.022 | 0.022 | 0.022 | 0.022 | 0.022 |

## G  ADDITIONAL VISUALIZATION RESULTS OF OUR ATTEXPLORE

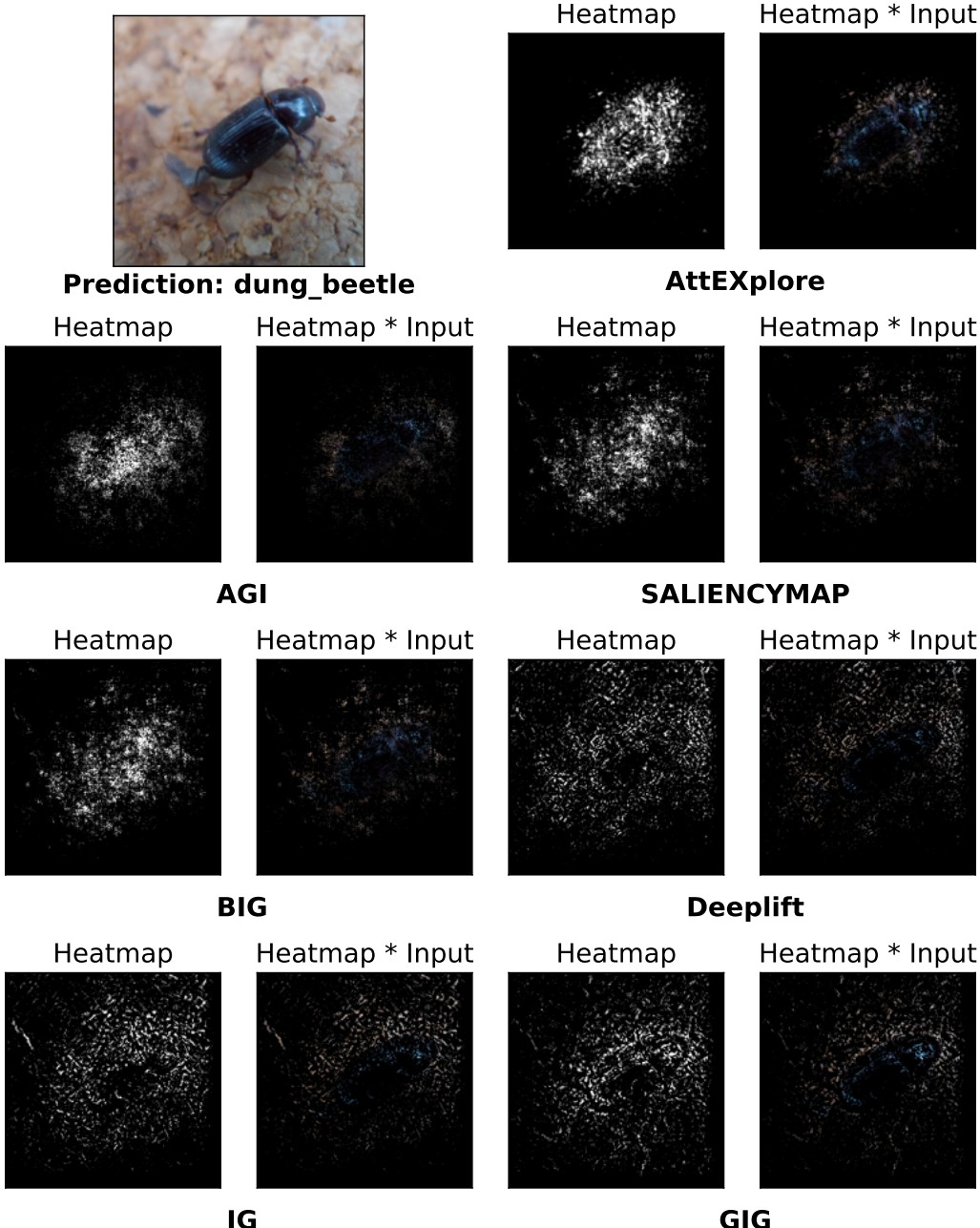

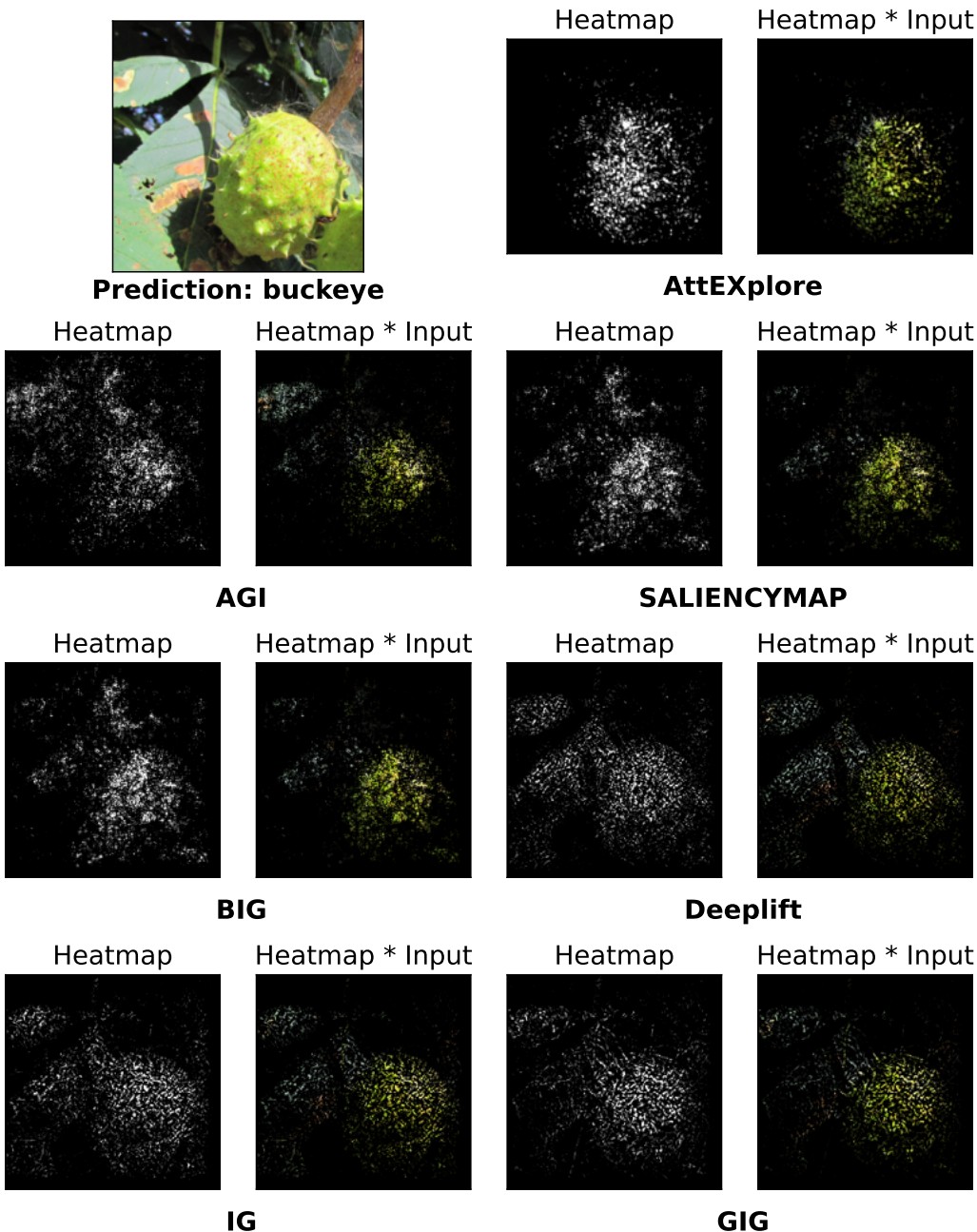

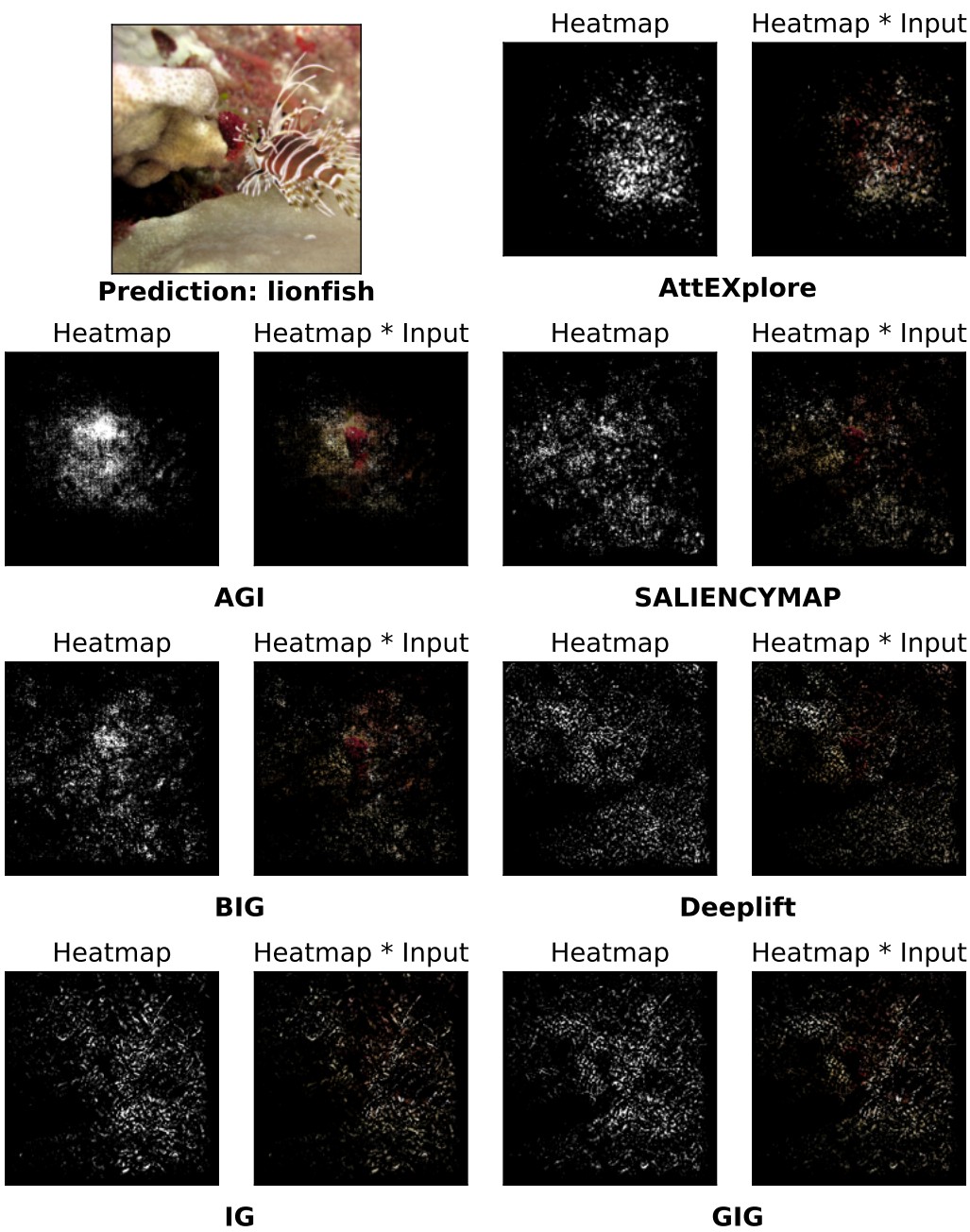

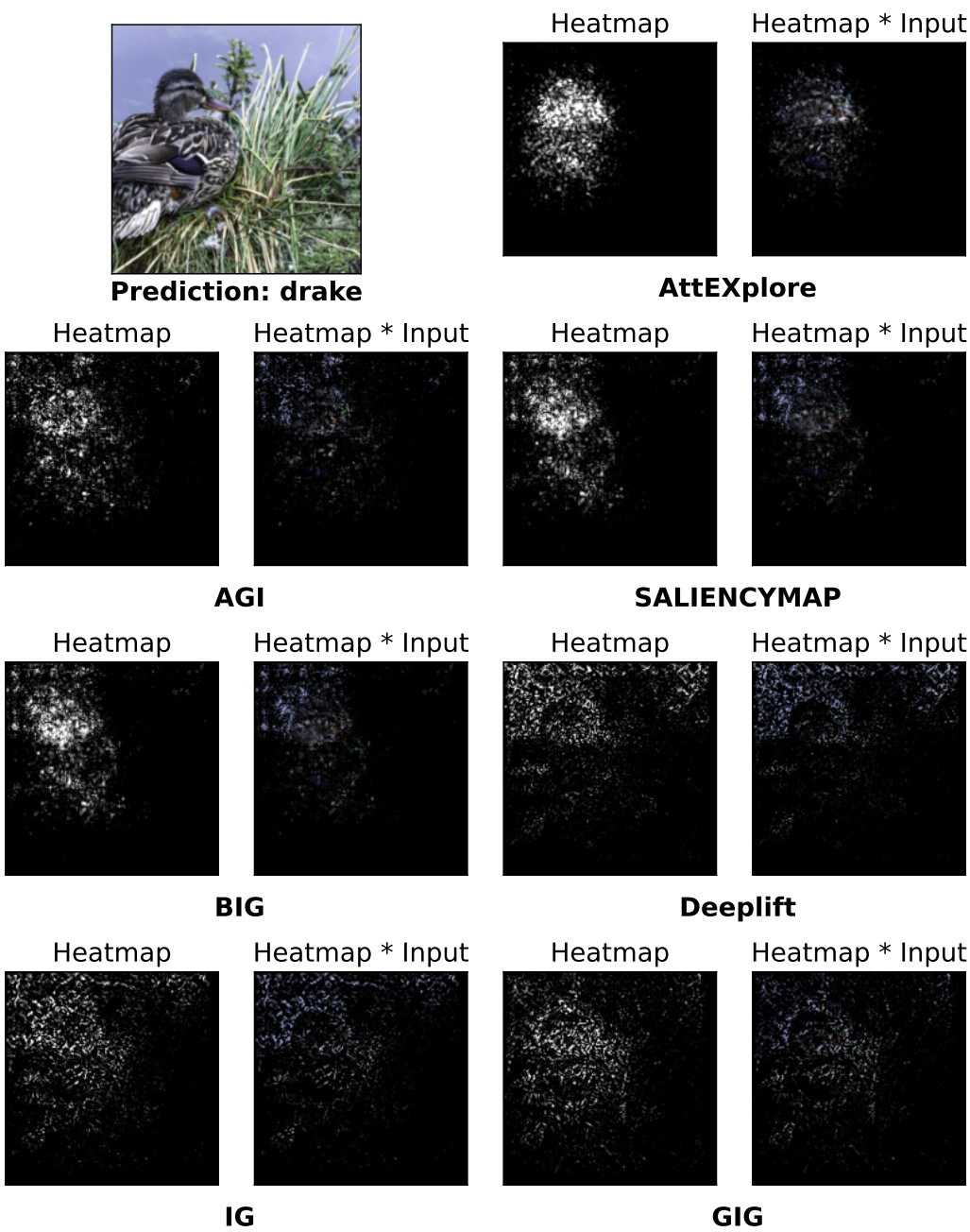

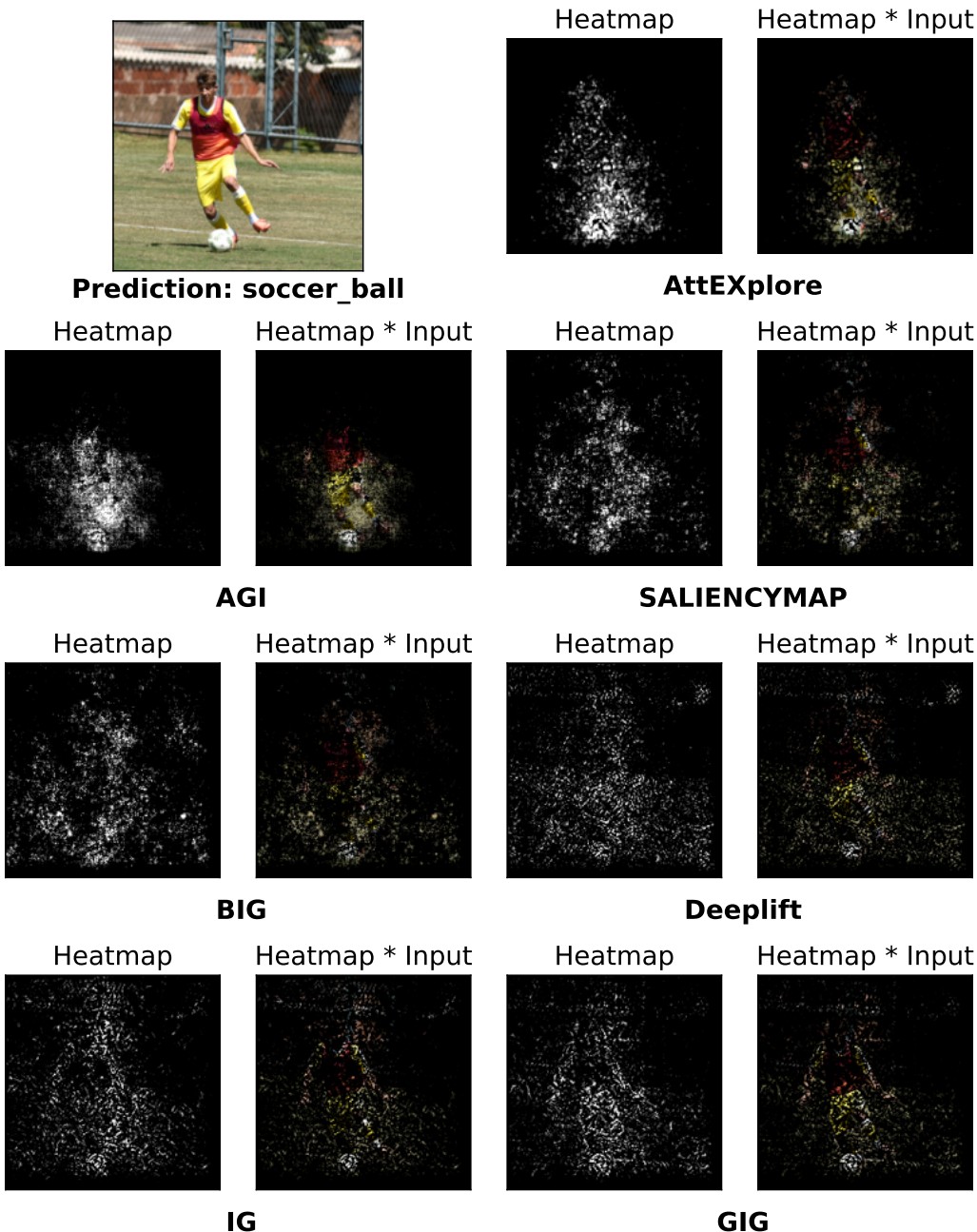

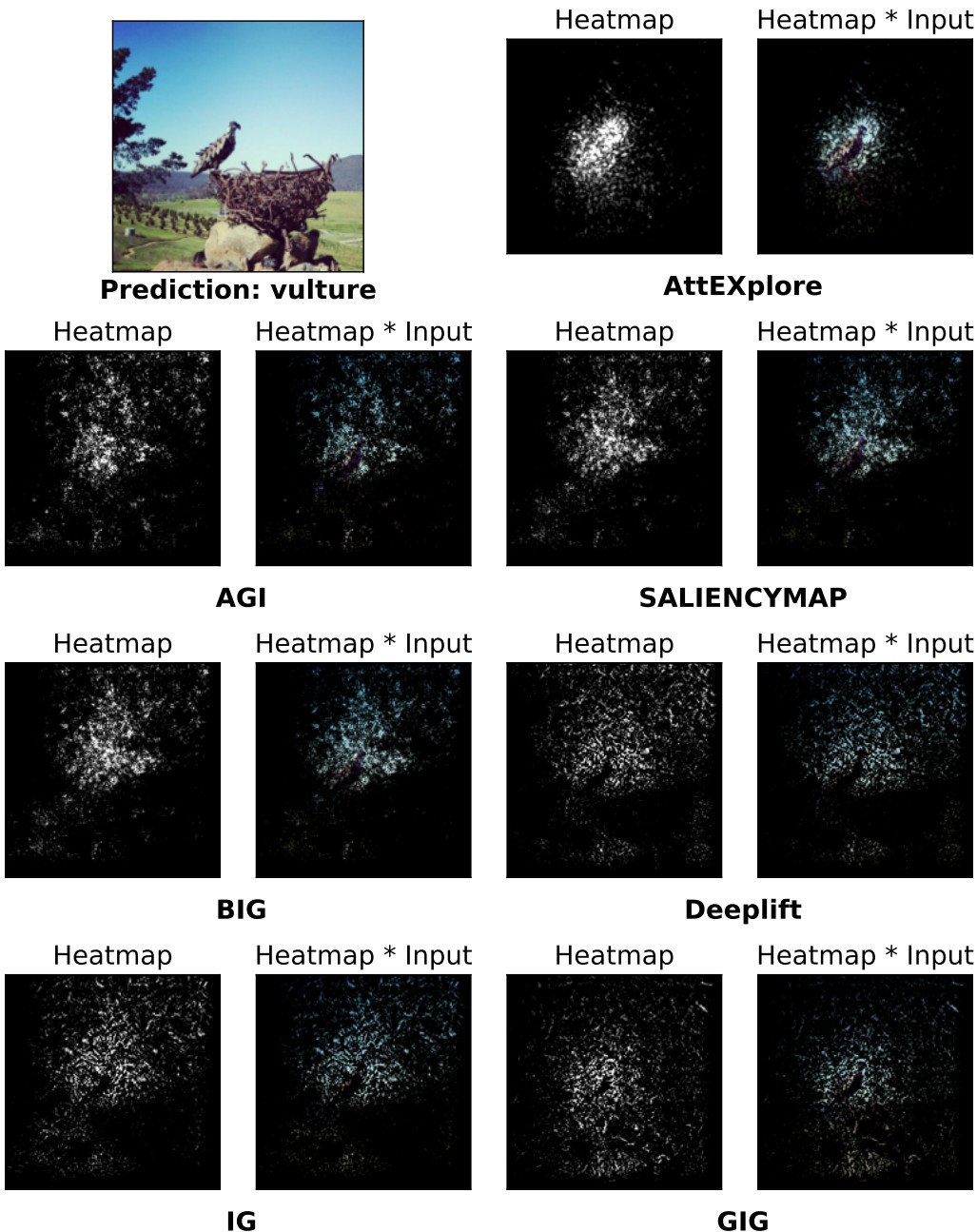

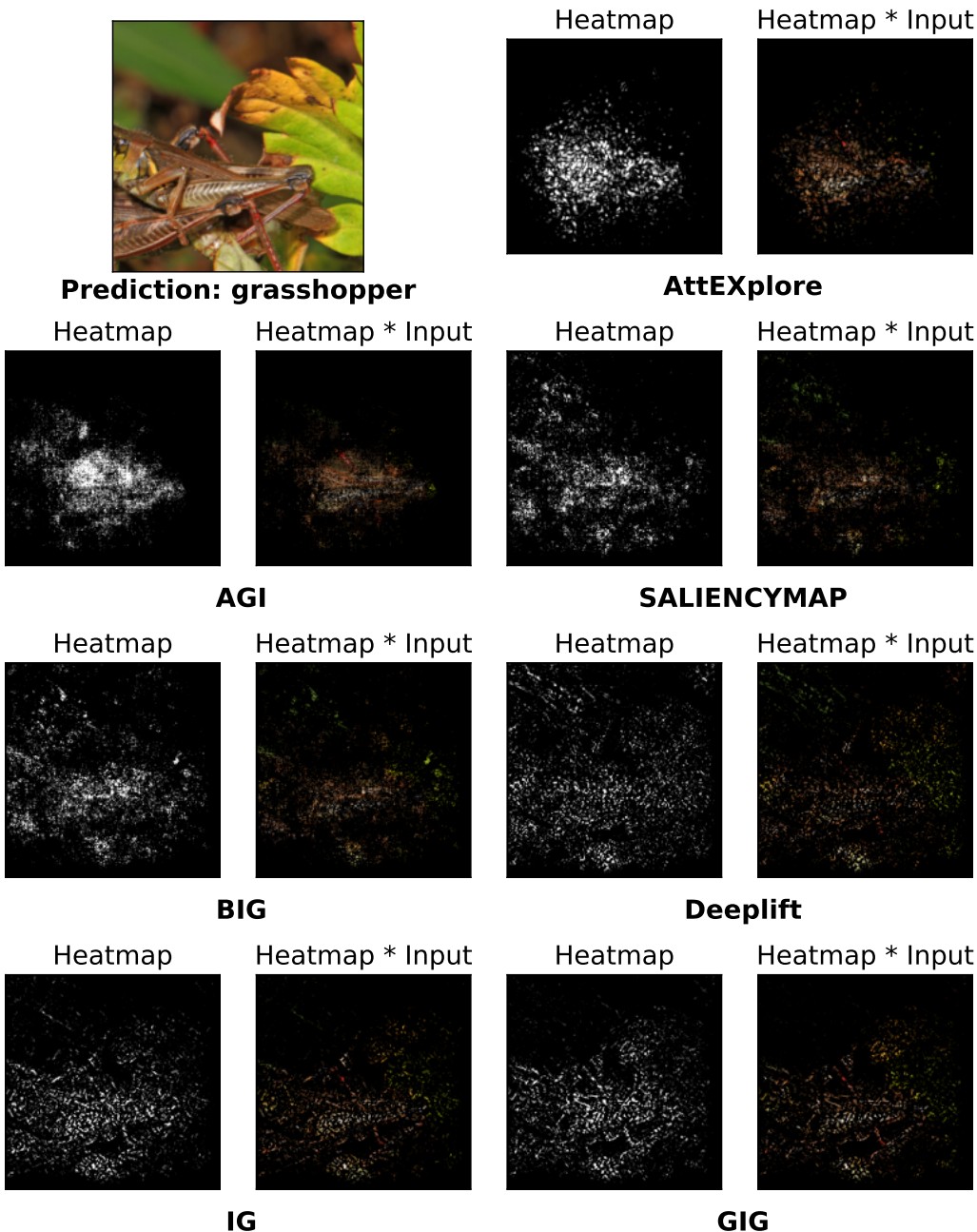

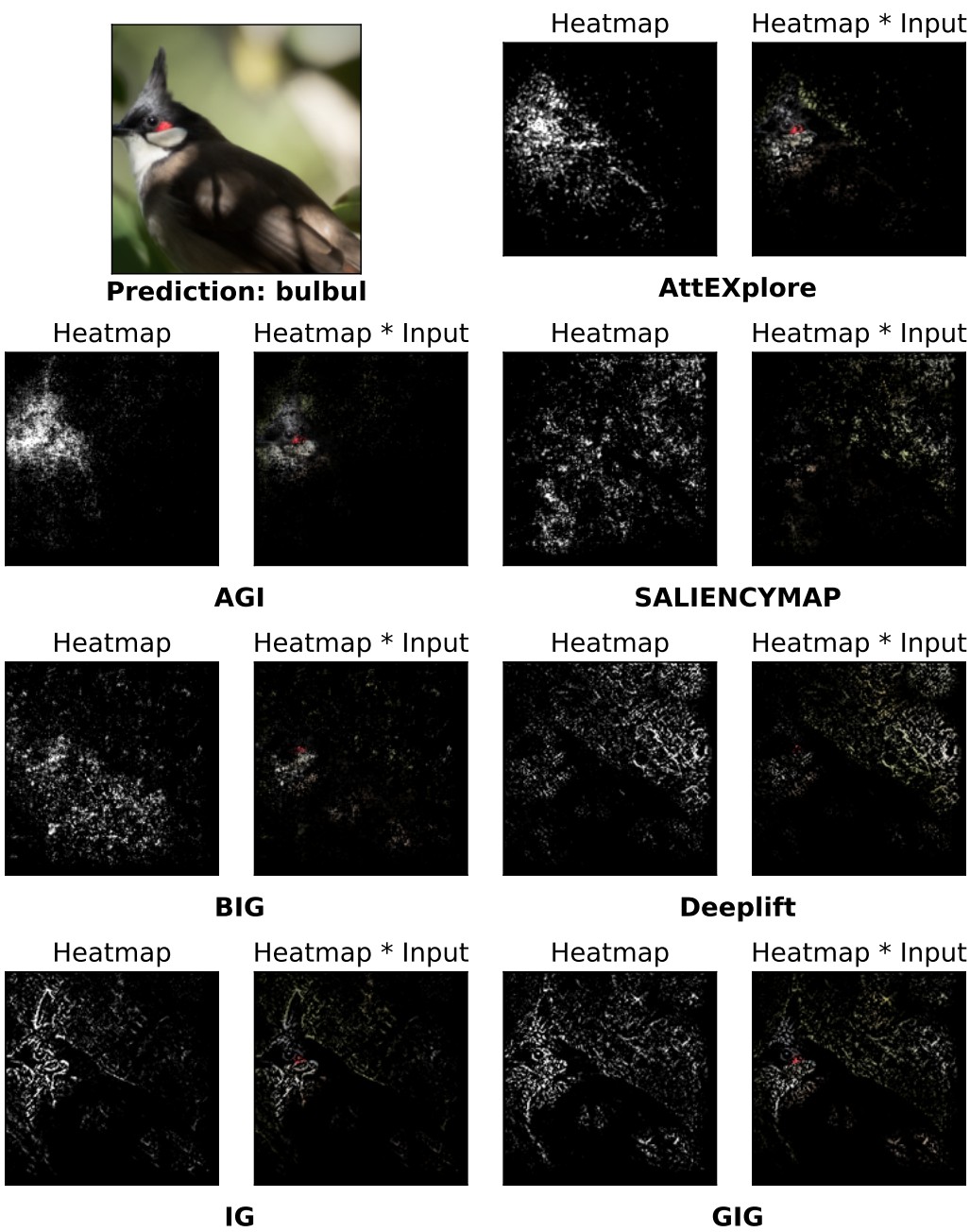

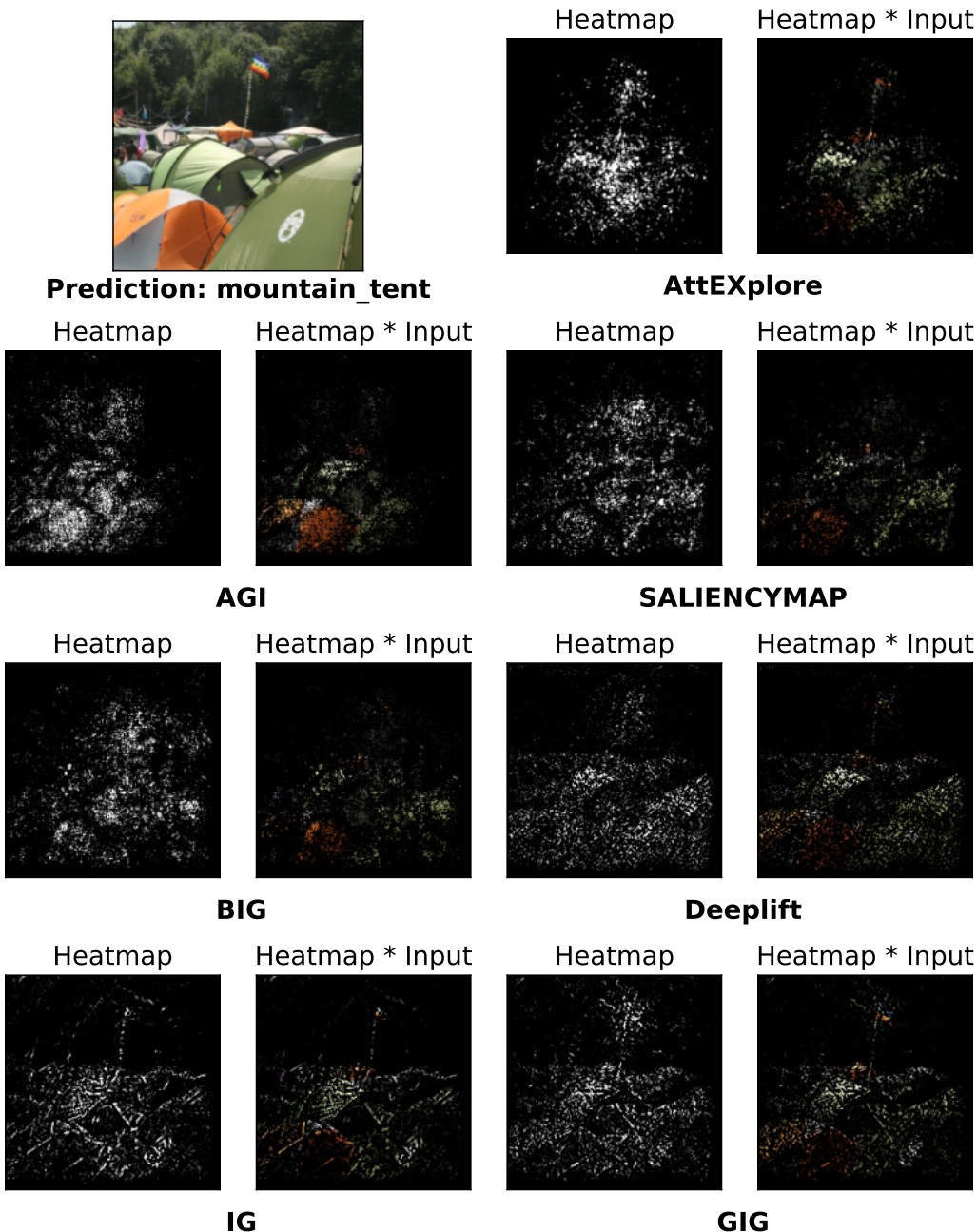

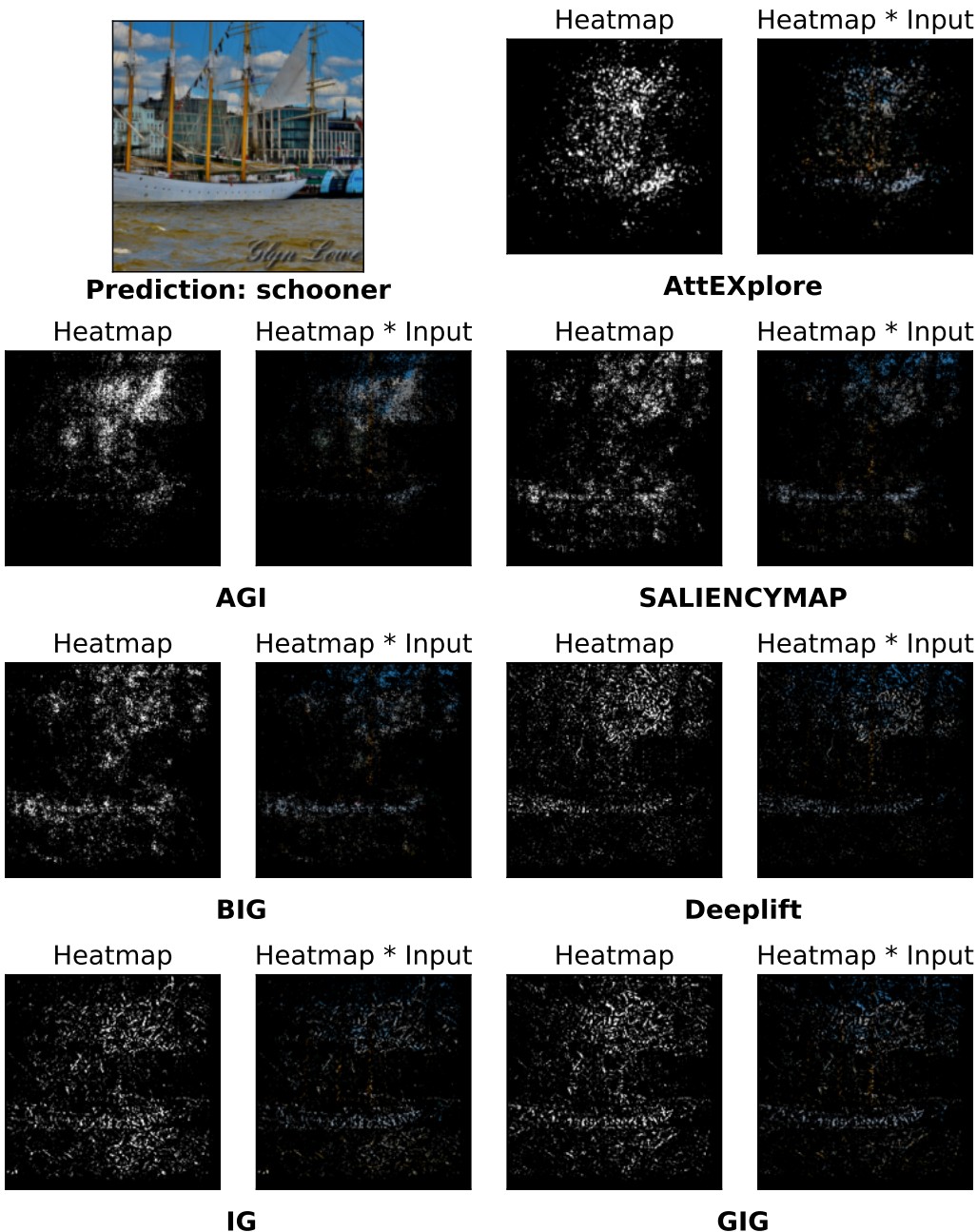

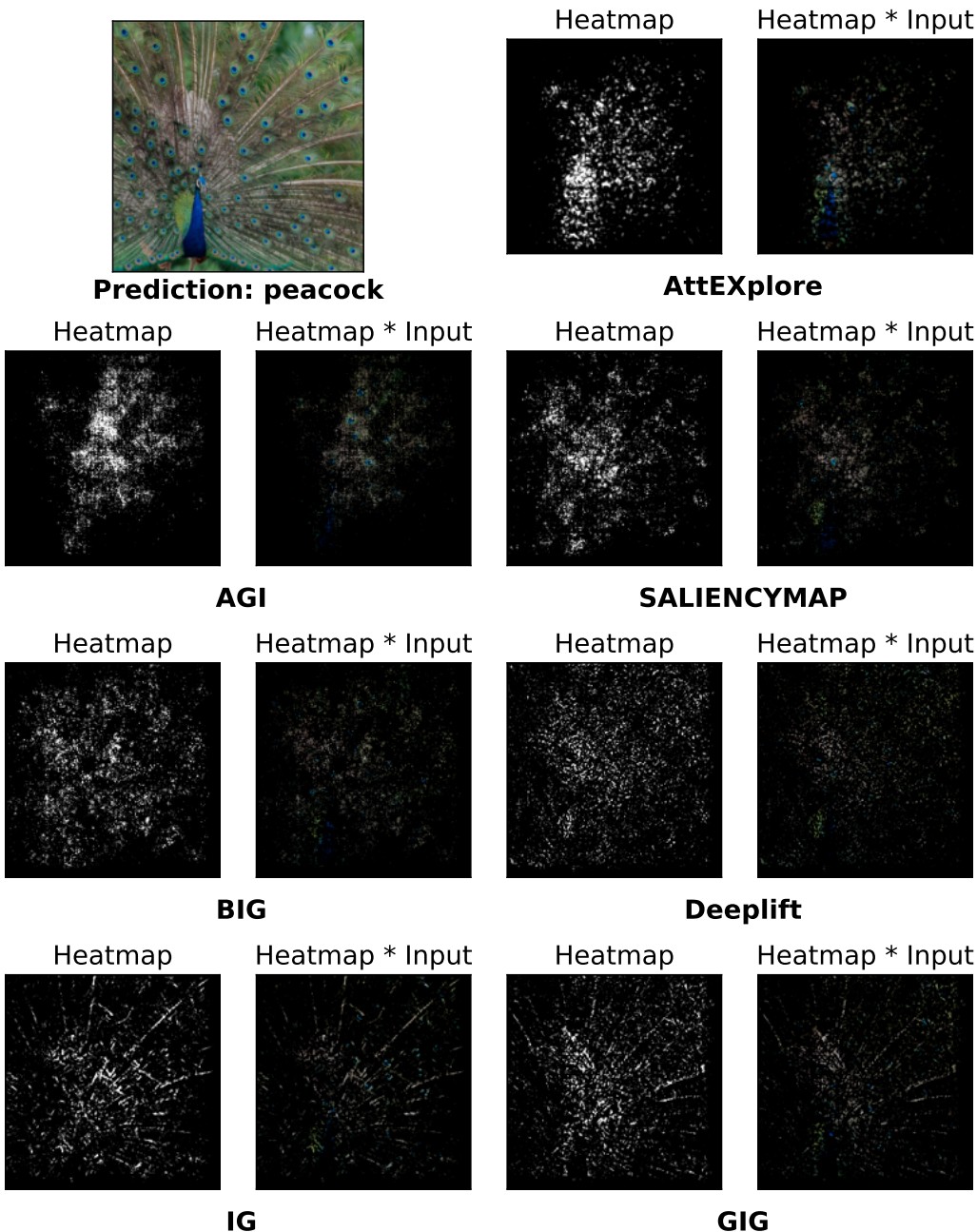

