# OpenReview forum: "AttEXplore: Attribution for Explanation with model parameters eXploration"
_ICLR.cc/2024/Conference — ICLR 2024 poster_

### Official Review · Reviewer_UxR1 · 2023-10-30

**Soundness:** 3 good
**Presentation:** 2 fair
**Contribution:** 2 fair
**Rating:** 6
**Confidence:** 3

**Summary:**

The paper proposes AttEXplore, an attribution framework based on transferable adversarial attacks. The authors claim that there are shared principles between the decision boundary exploration approaches of attribution and transferable attacks. Based on such observation, the framework performs attribution analysis based on the design of a nonlinear integration path. The frequency-based input feature alterations make the framework transferable across different decision boundaries.

**Strengths:**

1. The paper is well-written with a clear storyline that motivates the work.
2. The experiments on multiple attribution baselines are comprehensive.
3. The idea of bridging attribution methods and adversarial attacks is novel.

**Weaknesses:**

1. Given the transferable claim of this new attribution method, it will be necessary to show sufficient visualizations of how the attribution generalizes to different types of task models other than image classification.

2. The post-hoc explanation methods often suffer from false-positive responses, which highlight the semantic parts that are actually not relevant to the true grounding objects. The paper should elaborate more on how the framework overcomes such challenges. Similarly, the paper shall have more quantitative evaluations on whether the attribution is correctly aligning with the ground truth.

3. Recent years have witnessed the development of white-box transformers (e.g., [1]), whose self-attentions naturally emerge as attributions for the model decision. It remains a question as how will AttEXplore outperform these interpretable-by-design approaches.

[1] Yu et al., Emergence of Segmentation with Minimalistic White-Box Transformers.

**Questions:**

Please address my concerns listed in the weakness section. I look forward to the authors' response and I will possibly consider revising the rating based on the response.

---

> ### Author Response · Authors · 2023-11-15
> **Official reply to Reviewer UxR1**
>
> **Weaknesses:**
>
> 1. Thank you for the thoughtful consideration of our work. It is necessary to generalize our method to models for different tasks beyond image classification. Working on the interpretability of NLP models is a good direction. For example, verifying the interpretability of medical reports generated by applying NLP technology.
>      At present, our work is aimed at attribution interpretation of image related tasks, which is consistent with the current SOTA research works including AGI [1], BIG [2], GIG [3], and SG [4].
>
> 2. Given that our method satisfies the sensitivity and implementation invariance axioms in IG [5], this ensures a one-to-one correspondence between the model's input and output during attribution, guaranteeing the accuracy of the attribution process. We have provided detailed proofs of our method's adherence to these axioms in the appendix.
>
> 3. We appreciate this suggestion. However, attribution methods are model agnostic interpretability methods, which is different from the interpretable-by-design approaches for white-box transformers. While Vision Transformers (ViTs) represent a typical application of the Transformer model in computer vision tasks, we have additionally included the experiment result of our method on the ViT model. We anticipate the result in the global comments Table *4 may be helpful to further extend the comparison in the future to evaluate its interpretability performance in this domain. We will include the discussion with this specific white-box transformer [6] work in our revision.
>
> Reference:
>
> [1] Pan, D., Li, X., & Zhu, D. (2021, August). Explaining deep neural network models with adversarial gradient integration. In Thirtieth International Joint Conference on Artificial Intelligence (IJCAI).
>
> [2] Wang, Z., Fredrikson, M., & Datta, A. (2022). Robust models are more interpretable because attributions look normal. Proceedings of the 39th International Conference on Machine Learning.
>
> [3] Kapishnikov, A., Venugopalan, S., Avci, B., Wedin, B., Terry, M., & Bolukbasi, T. (2021). Guided integrated gradients: An adaptive path method for removing noise. In Proceedings of the IEEE/CVF conference on computer vision and pattern recognition (pp. 5050-5058).
>
> [4] Smilkov, D., Thorat, N., Kim, B., Viégas, F., & Wattenberg, M. (2017). Smoothgrad: removing noise by adding noise. the 34th International Conference on Machine Learning.
>
> [5] Sundararajan, M., Taly, A., & Yan, Q. (2017, July). Axiomatic attribution for deep networks. In International conference on machine learning (pp. 3319-3328). PMLR.
>
> [6] Yu, Y., Chu, T., Tong, S., Wu, Z., Pai, D., Buchanan, S. and Ma, Y., 2023. Emergence of segmentation with minimalistic white-box transformers. arXiv preprint arXiv:2308.16271.

---

> > ### Comment · Reviewer_UxR1 · 2023-11-22
> >
> > The response well addresses my concerns. I will rate this paper a borderline accept.

---

> > > ### Author Response · Authors · 2023-11-22
> > > **Thanks**
> > >
> > > Thank you for raising the score.

---

### Official Review · Reviewer_r7kS · 2023-11-01

**Soundness:** 2 fair
**Presentation:** 2 fair
**Contribution:** 3 good
**Rating:** 6
**Confidence:** 3

**Summary:**

The authors present AttEXplore, a novel feature attribution method for explaining Deep Neural Networks (DNNs), designed to address the growing need for transparent and interpretable AI models. AttEXplore is inspired by recent advances in the domain of transferable adversarial attacks, and it combines two essential components:

(1) Feature attributions based on adversarial gradient integration, and

(2) Frequency-based data augmentation techniques to enhance the robustness of explanations.

In their paper, the authors conduct an extensive evaluation of AttEXplore on the ImageNet dataset, employing three different DNN architectures (Inception-v3, ResNet-50, VGG-16). They compare AttEXplore's faithfulness and time complexity against other attribution methods, providing valuable insights into its performance. Furthermore, the authors perform an ablation study to better understand the impact of the key components within AttEXplore.

To facilitate the adoption of their method, the authors also provide the associated code, making it accessible for other researchers and practitioners seeking to utilize AttEXplore for their own applications.

**Strengths:**

The paper offers several notable strengths:

**Leveraging of Insights from Transferable Adversarial Attack Works:** The authors introduce a unique approach that draws inspiration from the field of transferable adversarial attacks. Particularly, their integration of frequency-based data augmentation techniques, aimed at reducing noise in heatmaps (usually caused by gradient shattering and/or high non-linearity of DNNs), is an innovative contribution. This demonstrates a thoughtful incorporation of knowledge from a related domain into the field of local XAI.

**Comprehensive Model Evaluation:** The authors carry out an extensive evaluation, involving multiple deep neural network models and the ImageNet dataset. This thorough examination of AttEXplore's performance across different DNN architectures enhances the credibility of their findings and demonstrates its versatility and applicability.

**Related Work Section:** The authors effectively contextualize their work within the existing literature, highlighting its relevance and significance in the broader research landscape.

**Open-Source Code Availability:** A notable strength of this paper is the provision of the code associated with AttEXplore. This openness enables other researchers and practitioners to readily adopt and build upon the proposed method, promoting further research and development in the area of local XAI."

**Weaknesses:**

The paper exhibits, however, several weaknesses:

**Clarity:**
Some sections of the paper lack clarity, making it challenging for readers to grasp the key concepts. The abstract and introduction suffer from being overly specific without providing sufficient context. Additionally, important elements of the proposed method are not adequately introduced, leaving readers without a clear understanding of the approach.

Figure 1, a critical element in conveying the method, could benefit from improved clarity. Enhancements to this figure would help readers better comprehend their motivation for AttEXplore.

Further, the paper frequently references the appendix but fails to provide specific locations. It would be highly beneficial if the authors included clear references to specific sections within the appendix, making it easier for readers to locate relevant supplementary information.

To me, some sentences in the paper do not make sense, such as "DeepLIFT and LIME generate explanations for generic models which limits the understanding of model’s global behaviors.“ Why would it limit the understanding, and what do you mean with global behvior?

In the Method section, Equation (4) includes index $i$ without a clear definition. It is unclear whether $I$ refers to a random draw of sample $x$ or a class index. This ambiguity needs to be resolved to enhance the precision of the method description.

**Evaluation Shortcomings:**
The evaluation has certain limitations. The absence of faithfulness curves is one deficiency as it hinders a comprehensive understanding of the method's performance. Moreover, the paper does not compare AttEXplore against state-of-the-art approaches designed to address noise in heatmaps, such as LRP (composite rules), SHAP, or LIME. Additionally, the time complexity comparison is limited to only a subset of methods, which restricts the scope of the evaluation and limits the ability to assess AttEXplore's competitive performance in terms of speed."

**Questions:**

Have you compared your method against other local XAI techniques that attempt to generate more robust explanations by reducing noisy attributions?

In the Method section, Equation (4) includes index $i$ without a clear definition. It is unclear whether $I$ refers to a random draw of sample $x$ or a class index. What does $i$ stand for?

---

> ### Author Response · Authors · 2023-11-15
> **Official reply to Reviewer r7kS (Part 1)**
>
> **Clarity:**
>
> Thanks for the constructive comment. We will rewrite the relevant sections of abstract and introduction to facilitate a more seamless understanding for readers. Additionally, we will improve Figure 1, enhancing image clarity and the representation of key elements to provide a clearer understanding of our method. We will also refine the structure of the appendix and provide more precise and specific references to its content within the main text for improved citation.
>
> Regarding the statement "DeepLIFT and LIME generate explanations for generic models which limits the understanding of the model’s global behaviors," our intention is to convey that DeepLIFT [1] and LIME [2], as local interpretability methods, primarily focus on explaining the local neighborhood behaviors of models. These methods typically employ simpler, interpretable models to approximate the behavior of complex models near specific data instances or attempt to perturb inputs to identify the inputs most responsible for a prediction in the neighborhood. Therefore, these two methods do not satisfy the sensitivity and implementation invariance axioms proposed by IG [3], resulting in limited global interpretability of model behaviors.  Global interpretability refers to understanding the entire behavior of the model, not just explanations for specific instances or local regions. Global interpretability focuses on properties such as features, weights, structure, and behavior patterns of the model across the entire input space.
>
> In Equation (4), the variable $i$ represents the number of frequency domain explorations, i.e., $i=1,2,...,N$. Ultimately, we generate $N$ adversarial samples through frequency domain exploration. We will review the main context to avoid similar ambiguities.

---

> > ### Comment · Reviewer_r7kS · 2023-11-22
> >
> > Thank you very much for the clarifications and your willingness to improve certain sections of your work.
> >
> > Unfortunately, I still can not follow your remark fully about DeepLIFT and LIME. First, because DeepLIFT is based on the (modified) gradient and thus does not require a simpler interpretable model or perturbs inputs. Also according to [3] (your reference), DeepLIFT fulfills the sensitivity axiom. Further, one could also apply LIME / or perturbation-based methods to weights and structures (however, might be computationally very demanding, as you also point out in your reply).

---

> ### Author Response · Authors · 2023-11-15
> **Official reply to Reviewer r7kS (Part 2)**
>
> **Evaluation Shortcomings:**
>
> Thanks for the reviewer's comments. Our method satisfies the Completeness axiom, as proven in Section 3 of IG [3], serving as a sanity check. We have also conducted INFD score tests [4] in the global comments in Table *2, and the results demonstrate that our method exhibits superior robustness and fidelity.
>
> Concerning LIME [2], LRP [5], and SHAP [6], we would like to note that these methods have different requirements compared to current state-of-the-art attribution methods.
> Firstly, LIME attempts to perturb inputs to identify the inputs most responsible for a prediction in the neighborhood. In addition, LIME’s interpretable behavior requires cluster segmentation of images, so its interpretation model is not point-to-point. And if it performs point-to-point calculations, the computational complexity of LIME will be extremely high. LRP relies on model structure and is challenging to implement. DeepLIFT is a more general case of LRP, and our method surpasses DeepLIFT. SHAP, when applied to samples with high dimensions, uses approximate results for inference and involves a significant amount of interaction operations, resulting in extremely high computational complexity.
>
> Secondly, IG [3] extensively discusses that these methods do not necessarily satisfy the sensitivity and implementation invariance axioms. For example, LRP uses discrete gradients to compute feature importance, which does not satisfy the sensitivity axiom and may lead to instances where attributions change, but the attribution result is 0, rendering interpretability completely ineffective.
>
> Lastly, current state-of-the-art methods including AGI [7], BIG [8] and GIG [9] don’t include LIME, LRP, and SHAP as the comparison in their experiments. It is noteworthy that some of the methods are noise reduction-based methods which were published much earlier. In our baseline, GIG represents the state-of-the-art noise reduction-based method as proposed in 2021. We have achieved significant improvement over the GIG method.
>
> Regarding the subset selection for time complexity comparison, we selected the five methods that are closest in performance to our method for efficiency comparison. Other methods, such as SaliencyMap, DeepLIFT and EG have relatively poor attribution accuracy in the baseline, making them unnecessary for comparison. Although Fast-IG achieves faster attribution speeds, its accuracy is unfortunately the lowest.
>
> Reference:
>
> [1] Shrikumar, A., Greenside, P., & Kundaje, A. (2017, July). Learning important features through propagating activation differences. In International conference on machine learning (pp. 3145-3153). PMLR.
>
> [2] Ribeiro, M. T., Singh, S., & Guestrin, C. (2016, August). " Why should i trust you?" Explaining the predictions of any classifier. In Proceedings of the 22nd ACM SIGKDD international conference on knowledge discovery and data mining (pp. 1135-1144).
>
> [3] Sundararajan, M., Taly, A., & Yan, Q. (2017, July). Axiomatic attribution for deep networks. In International conference on machine learning (pp. 3319-3328). PMLR.
>
> [4] Yeh, C. K., Hsieh, C. Y., Suggala, A., Inouye, D. I., & Ravikumar, P. K. (2019). On the (in) fidelity and sensitivity of explanations. Advances in Neural Information Processing Systems, 32.
> [5] Bach, S., Binder, A., Montavon, G., Klauschen, F., Müller, K. R., & Samek, W. (2015). On pixel-wise explanations for non-linear classifier decisions by layer-wise relevance propagation. PloS one, 10(7), e0130140.
>
> [6] Lundberg, S. M., & Lee, S. I. (2017). A unified approach to interpreting model predictions. Advances in neural information processing systems, 30.
>
> [7] Pan, D., Li, X., & Zhu, D. (2021, August). Explaining deep neural network models with adversarial gradient integration. In Thirtieth International Joint Conference on Artificial Intelligence (IJCAI).
>
> [8] Wang, Z., Fredrikson, M., & Datta, A. (2022). Robust models are more interpretable because attributions look normal. Proceedings of the 39th International Conference on Machine Learning.
>
> [9] Kapishnikov, A., Venugopalan, S., Avci, B., Wedin, B., Terry, M., & Bolukbasi, T. (2021). Guided integrated gradients: An adaptive path method for removing noise. In Proceedings of the IEEE/CVF conference on computer vision and pattern recognition (pp. 5050-5058).

---

> > ### Comment · Reviewer_r7kS · 2023-11-22
> >
> > I agree that LIME and SHAP are infeasible to apply on a pixel-level level. However, still, my point remains that it would be of interest to compare your method against explanation methods that result in less noisy heatmaps. This also includes GradCAM, which [1] includes in the evaluation.
> >
> > Regarding the complexity comparison: is SaliencyMap [2] not simply the gradient (i.e., derivative of output of softmax class node with respect to the input image), thus computationally very efficient? I still think it will be valuable to include all evaluated methods to understand the trade-off between attribution faithfulness/accuracy and efficiency.
> >
> > References:
> > [1] Andrei Kapishnikov, Subhashini Venugopalan, Besim Avci, Ben Wedin, Michael Terry, Tolga Bolukbasi:
> > Guided Integrated Gradients: An Adaptive Path Method for Removing Noise. CVPR 2021: 5050-5058
> >
> > [2] Karen Simonyan, Andrea Vedaldi, Andrew Zisserman: Deep Inside Convolutional Networks: Visualising Image Classification Models and Saliency Maps. ICLR (Workshop Poster) 2014

---

> ### Author Response · Authors · 2023-11-22
> **Our clarification (Part 1)**
>
> Thanks to the reviewer for the response. We hope that the following detailed discussion will clear up your confusion:
>
> First of all, we would like to correct our point of view. In fact, DeepLIFT [1] does not all satisfy the two axioms proposed by IG [2]. Although DeepLIFT does not require a simpler interpretable model or perturbs inputs, and satisfies the sensitivity axiom, DeepLIFT breaks Implementation Invariance axiom. The corresponding original text in IG is at the beginning of Section 2: **We now discuss two axioms (desirable characteristics) for attribution methods. We find that other feature attribution methods in literature break at least one of the two axioms. These methods include DeepLift ( Shrikumar et al., 2016; 2017), Layer-wise relevance propagation (LRP) (Binder et al., 2016), Deconvolutional networks (Zeiler & Fergus, 2014), and Guided back-propagation (Springenberg et al., 2014) .**
>
> A more direct explanation can be found in the second paragraph of Section 2.2: **We now discuss intuition for why DeepLift and LRP break Implementation Invariance.**
>
> Our attribution method satisfies the two axioms of sensitivity and implementation invariance, which enables one-to-one correspondence between model input and output to obtain more accurate attribution results.
>
> Secondly, gradient-based and perturbation-based methods are currently two important directions in interpretability methods. LIME [3] or other perturbation-based methods obtain the contribution of the model input to the output through perturbation. But this is not consistent in principle with our gradient-based attribution method, where we leverage the backward pass of a neural network to assess the influence and relevance of an input feature on the model decision. In addition, as you said, LIME requires a large amount of calculation, which is one of its disadvantages.
>
>
> [1] Shrikumar, A., Greenside, P., & Kundaje, A. (2017, July). Learning important features through propagating activation differences. In International conference on machine learning (pp. 3145-3153). PMLR.
>
> [2] Sundararajan, M., Taly, A., & Yan, Q. (2017, July). Axiomatic attribution for deep networks. In International conference on machine learning (pp. 3319-3328). PMLR.
>
> [3] Ribeiro, M. T., Singh, S., & Guestrin, C. (2016, August). " Why should i trust you?" Explaining the predictions of any classifier. In Proceedings of the 22nd ACM SIGKDD international conference on knowledge discovery and data mining (pp. 1135-1144).

---

> ### Author Response · Authors · 2023-11-22
> **Our clarification (Part 2)**
>
> Thank you for your constructive comments. GradCAM [1] needs to select a layer in the middle (the dimension is lower than the original image dimension and has multiple channels). Feature channels are combined based on the gradient sum of each channel. Because of the low dimensionality, pixel-level attribution cannot be achieved. In addition, since it only uses the gradient of a single state, similar to Saliency Map [2], it does not satisfy Sensitivity. Although we believe that GradCAM is not an attribution algorithm in the strict sense (does not satisfy the definition and axioms of attribution) and it cannot be on the same track as our method, we are happy to provide the comparison results of GradCAM. While we believe this is not necessarily fair, our method still perform better than GradCAM. Because our method is better than GradCAM in both insertion score and deletion score.
>
> In order to prove the effect of our method on other models, we also conducted additional experiments on the currently widely used Vision Transformer (ViT) models. The experimental results can be seen in Table 4 in the global comment. Our method can still achieve the best results on the ViT models.
>
> **We will update all details of this experiment to the anonymous link of the code.**
>
> We apologize for our honest fault in the rebuttal reply in Official reply to Reviewer r7kS (Part 2), which is a typo for 'Other methods, such as SaliencyMap, involve extensive computations, making them slower'. In fact, as you point out, Saliency Map is more efficient. We would like to clarify that **Other methods, such as SaliencyMap, DeepLIFT [3] and EG [4] have relatively poor attribution accuracy in the baseline, making them unnecessary for comparison.** We will correct this typo in the original reply.
>
> We have listed the speed analysis table of the algorithm in Section 5.3 of the article. It can be seen that the efficiency of our algorithm has achieved the optimal effect in the IG-based attribution method. On the other hand, we believe that the faithfulness of attribution is more important than the efficiency of the interpretability.
>
> | 　         | Inception-v3 |          | ResNet-50 |          | VGG-16    |          |
> |------------|--------------|----------|-----------|----------|-----------|----------|
> | Method     | Insertion    | Deletion | Insertion | Deletion | Insertion | Deletion |
> | GradCAM    | 0.4496       | 0.1084   | 0.2541    | 0.0942   | 0.3169    | 0.0841   |
> | AttEXplore | 0.4732       | 0.0297   | 0.4197    | 0.0293   | 0.3186    | 0.0226   |
>
> [1] Selvaraju, R. R., Cogswell, M., Das, A., Vedantam, R., Parikh, D., & Batra, D. (2017). Grad-cam: Visual explanations from deep networks via gradient-based localization. In Proceedings of the IEEE international conference on computer vision (pp. 618-626).
>
> [2] Karen Simonyan, Andrea Vedaldi, Andrew Zisserman: Deep Inside Convolutional Networks: Visualising Image Classification Models and Saliency Maps. ICLR (Workshop Poster) 2014
>
> [3] Shrikumar, A., Greenside, P., & Kundaje, A. (2017, July). Learning important features through propagating activation differences. In International conference on machine learning (pp. 3145-3153). PMLR.
>
> [4] Erion, G., Janizek, J. D., Sturmfels, P., Lundberg, S. M., & Lee, S. I. (2021). Improving performance of deep learning models with axiomatic attribution priors and expected gradients. Nature machine intelligence, 3(7), 620-631.

---

> > ### Comment · Reviewer_r7kS · 2023-11-23
> >
> > Thank you very much again for your additional results and detailed clarifications.
> > With these additional results for the evaluation section and a rewriting to improve clarity of some sections in the main manuscript, I am increasing my score from 5 to 6 (marginally above the acceptance threshold).

---

> > > ### Author Response · Authors · 2023-11-23
> > > **Thanks**
> > >
> > > Thank you very much for raising the score.

---

### Official Review · Reviewer_M36V · 2023-11-01

**Soundness:** 3 good
**Presentation:** 3 good
**Contribution:** 3 good
**Rating:** 6
**Confidence:** 4

**Summary:**

This paper introduces a novel attribution method called AttEXplore, designed for explaining DNN models through the exploration of model parameters. AttEXplore leverages the concept of transferable attacks as there's consistency between the decision boundary exploration approaches of attributionand the process for transferable adversarial attacks. By manipulating input features according to their frequency information, AttEXplore enhances the interpretability of DNN models.

**Strengths:**

1. This paper is well-written and easy to follow.
2. AttEXplore outperforms existing methods on the ImageNet dataset and models such as Inception-v3, ResNet-50, and VGG-16, achieving higher insertion scores and lower deletion scores, which underscores its effectiveness in model evaluation.
3. AttEXplore exhibits superior computational efficiency in terms of the number of frames processed per second when compared to existing methods.

**Weaknesses:**

1. This paper appears to focus exclusively on evaluating AttEXplore using image data, it lacks experiments on other data modalities, such as text data on NLP models. Expanding the evaluation to different data modalities could provide a more comprehensive assessment of AttEXplore's applicability.
2. For image classification models, the authors did experiments on CNN model groups including Inception-v3, ResNet-50, and VGG-16. It is unclear that if the superior explainability of AttEXplore stands still on other model groups, like vision transformers and MLP model groups. Extending the evaluation to diverse model architectures would further validate its effectiveness across different model types.

**Questions:**

It's shown in Table 1 in the appendix that compared with N = 10, the drop of insertion score  for N = 1is not huge (~0.007 for inception-v3, still better than existing methods ). The perturbation rate was set as 16, while the model performance is better when perturbation rate is 48. Would the trend of model performance under different N be more clear when the perturbation rate is set as a larger value?

---

> ### Author Response · Authors · 2023-11-15
> **Official reply to Reviewer M36V**
>
> **Weaknesses:**
>
> 1. Thanks for pointing out the valuable comment for attribution method research. Actually, exploring interpretability in the context of NLP models is undoubtedly a valuable direction, especially in domains such as medical report generation where interpretability is crucial.
> Currently, our work focuses on attribution interpretation for images, aligning with the current state-of-the-art approaches like AGI [1], BIG [2], GIG [3], SG [4], and other baselines. We would like to provide a discussion of potential work following the comment for NLP works.
> 2. We appreciate the reviewer's constructive suggestions. We have conducted the additional experiment on the model of Vision Transformers (ViTs). ViTs represent an excellent vision model based on the Transformer architecture. Different from traditional DNN models, ViTs process images as sequences of patches, leading to a patch-like interpretation of features [5]. The experimental results on ViTs, included in the global comments Table *4, still demonstrate that our method achieves the best performance.
>
> **Questions:**
>
> We are afraid the answer is negative for the trend of model performance under different $N$, even with a larger perturbation rate.
>
> On one hand, the ablation experiments in Tables 3, 4 and 5, and the appendix file, indicate that increasing the number of generated features or perturbation rates, among other hyperparameters, can enhance attribution accuracy to some extent. This is rational as the increase in the number of generated features ($N$) implies the exploration of more frequency domain information, while an increase in $\epsilon$ signifies the generation of higher-quality adversarial samples. Consequently, the performance at a perturbation rate of 48 may surpass the result with a perturbation rate of 16.
>
> On the other hand, the impact of altering hyperparameters on performance, as observed in ablation experiments, is relatively small. This suggests that the improvement of our method does not heavily rely on the specific choice of hyperparameters. However, when the perturbation rate is set to a larger value such as 48, the trend in model performance across different $N$ values does not become more clear. This might be attributed to the fact that a larger perturbation rate represents a larger search space. Although the number of explored samples increases, it doesn’t mean that all these samples are necessarily effective for the attribution result. We have validated this observation in the additional experiments provided in the global comments Table *5.
>
> Reference:
>
> [1] Pan, D., Li, X., & Zhu, D. (2021, August). Explaining deep neural network models with adversarial gradient integration. In Thirtieth International Joint Conference on Artificial Intelligence (IJCAI).
>
> [2] Wang, Z., Fredrikson, M., & Datta, A. (2022). Robust models are more interpretable because attributions look normal. Proceedings of the 39th International Conference on Machine Learning.
>
> [3] Kapishnikov, A., Venugopalan, S., Avci, B., Wedin, B., Terry, M., & Bolukbasi, T. (2021). Guided integrated gradients: An adaptive path method for removing noise. In Proceedings of the IEEE/CVF conference on computer vision and pattern recognition (pp. 5050-5058).
>
> [4] Smilkov, D., Thorat, N., Kim, B., Viégas, F., & Wattenberg, M. (2017). Smoothgrad: removing noise by adding noise. arXiv preprint arXiv:1706.03825.
>
> [5] Ma, W., Li, Y., Jia, X., & Xu, W. (2023). Transferable Adversarial Attack for Both Vision Transformers and Convolutional Networks via Momentum Integrated Gradients. In Proceedings of the IEEE/CVF International Conference on Computer Vision (pp. 4630-4639).

---

### Official Review · Reviewer_H27T · 2023-11-02

**Soundness:** 2 fair
**Presentation:** 2 fair
**Contribution:** 2 fair
**Rating:** 5
**Confidence:** 3

**Summary:**

This paper introduces a new attribution technology AttExplore that improves over three existing gradient integral methods, IG, BIG and AGI. The main contributions are that 1) the proposed method finds a baseline in the integral that serves as an adversarial input for the current model and some other variations; 2) their empirical results show improved insertion and deleting scores compared to baselines.

**Strengths:**

Authors have made it clear how AttExplore computes the attribution scores. Their experiments include 3 networks and a rich amount of baselines. Table 1 shows a good amount of improvement over the best baseline.

**Weaknesses:**

My main concerns include the motivation and the evaluation of this paper.

### Motivation
I am not entirely sure I understand the motivation of the paper. It looks like this paper accuses BIG and AGI of using the exact boundary of the underlying model, which fails to be generic to similar models with variations in the decision boundary? This is pretty counterintuitive to me as feature attributions are often perceived as local explanations at the given model and the given input. Why do we need to consider if the explanation generalizes other models? With that being said, generalizing to other decision boundaries may trade in the faithfulness of the underlying model (this would relate to my second question about faithfulness and I will explain later).

### Generalization of Frequency-based Methods?
The motivation to alter features in the frequency domain is because there are works showing high-frequency features help the model to generalize better [1]. I am pretty worried the authors explain this conclusion as altering some features in the frequency domain helps to create examples that transfer better, especially for methods just proposed in this paper (Eq. 4 - 5).  Have you verified your adversarial examples actually transfer better? Can you provide some analytical results convincing me this helps better explore more decision boundaries? So far all descriptions are pretty hand-wavy.

### Evaluations
Is the method faithful to the underlying model? It looks like to realize “a more general adversarial input” the proposed method manages to find an adversarial example that is much further to the decision boundary compared to the previous methods. I think authors may want to compare how much farther you go. It is pretty concerning to me that no matter how you adjust the feature generations in Table 3 and steps in Table 4, the proposed method has almost identical results on your metrics. Similar, no matter how you decrease $\epsilon$ in Table 5, the results do not change at all. What should we make out of these results? Are they showing the proposed method may be actually unfaithful to something? BTW, using $\epsilon=8$ is pretty huge and I think decision boundaries are usually much much closer than $8/255$ for models without adversarial training.

I recommend testing INFD score [2] and run sanity check [3].

### Ending

Unlike doing detection or being more robust, I really think research works in explaining feature importance are not result-driven. Namely, it is not about being state-of-the-art on some scores. Not every attribution method uses deletion and inserting scores in baseline papers cited. There are a lot of discussions for the unreliability of feature attributions that are not cited here. It is fine to say “we do not care about those critiques but just want to improve the current method” but it is important to point out *important flaws* in the existing methods and fix those.

[1] Haohan Wang, Xindi Wu, Zeyi Huang, and Eric P Xing. High-frequency component helps explain the generalization of convolutional neural networks. In Proceedings of the IEEE/CVF conference on computer vision and pattern recognition, pp. 8684–8694, 2020b.

[2] Yeh, C. K., Hsieh, C. Y., Suggala, A., Inouye, D. I., & Ravikumar, P. K. (2019). On the (in) fidelity and sensitivity of explanations. Advances in Neural Information Processing Systems, 32.

[3] Adebayo, J., Gilmer, J., Muelly, M., Goodfellow, I., Hardt, M., & Kim, B. (2018). Sanity checks for saliency maps. Advances in neural information processing systems, 31.

**Questions:**

N/A

---

> ### Author Response · Authors · 2023-11-15
> **Official reply to Reviewer H27T (Part 1)**
>
> **Motivation:**
>
> We appreciate the comments about the motivation and are sorry about the confusion.
>
> We would like to clarify a misunderstanding about the comparison with BIG and AGI (which are the SOTA methods), by highlighting the novel contribution which is also noted by other reviewers. The technical contribution is actually to leverage the insight from transferable adversarial attacks.
>
> The motivation is to enhance the attribution performance by considering different decision boundaries with the insights from transferable adversarial attacks. We consider the features to be crucial for the model output if the altered sample will cross the decision boundary. The summarization of the contributions of these features is considered as the attribution results. However, the training process from literature methods is limited to its capability of finding the accurate decision boundary (most variations in the decision boundary are OOD samples). How to effectively identify and leverage the decision boundary is challenging.
>
> Thus, inspired by the exploration of decision boundaries in transferable attacks, we observe the consistency of such adversarial sample generation with the samples for attribution. The reason is that a transferable adversarial sample on a given model also needs multiple operations of exploring the decision boundaries for a robust result. In this way, we consider the obtained sample will be able to cross more decision boundaries without the parameters from the target model, which means a more accurate attribution result.
> More discussion is available in Section 4.1, where we provide the reasons about the possibility of utilizing the samples to explore the decision boundary during the training process.
>
> Moreover, we argue that the performance of current attribution methods is generally influenced by the choice of different baselines. When the underlying task is complex, selecting an appropriate baseline is often difficult and ad-hoc. Therefore, for the decision boundaries of specific models corresponding to different underlying tasks, the local explanation of IG may not be optimal. For BIG, it uses a linear attribution path based on adversarial samples, but it cannot effectively solve this problem because linear attribution paths are unstable when crossing the decision boundaries of nonlinear complex models. For AGI, although it incorporates adversarial samples and nonlinear attribution paths other than the subjective selection of baselines, it aims at targeted adversarial attack, which may cover some unnecessary decision boundaries before successfully exploring the decision boundary corresponding to the target label, potentially leading to overfitting issues.
>
> In conclusion, to obtain a decision boundary that allows samples to cross without overfitting concern, we consider directly generating multiple decision boundaries with slight offsets through parameter adjustment is not practical and effective. Therefore, we conduct input exploration with transferable adversarial attack to simulate the process of sample generation by crossing the decision boundaries, thereby obtaining more accurate attribution.
>
> **Generalization of Frequency-based Methods?**
>
> Please see the analytical results provided in the global official comment in Table *1-1 and *1-2, which shows the result of attack success rate with different transferable adversarial attack methods. The generated adversarial samples are tested for the attack success rate on other victim models, in which a higher attack success rate represents a higher potential for crossing other general decision boundaries [1]. We extend our discussion of the motivation to leverage the insights from transferable attacks for decision boundaries exploration to further enhance the attribution method performance. In practice, it means that, if the adversarial samples can effectively cross various decision boundaries, it implies a higher transferability for transferable attacks and, for attribution, a better capability to obtain more accurate identification of critical features. Additionally, we emphasize that our findings extend beyond frequency domain transformations. We have tested the impact of different input transformations for interpretability.

---

> ### Author Response · Authors · 2023-11-15
> **Official reply to Reviewer H27T (Part 2)**
>
> **Evaluations:**
>
> Regarding Tables 3, 4 and 5, these results are for the ablation study, which is to evaluate the impact of different hyperparameters. The conclusion is that, by increasing the number of generated features or perturbation rates, the attribution accuracy may be slightly increased. This is also apparent since an increase in $N$ implies the exploration of more frequency domain information, and an increase in $\epsilon$ signifies higher-quality adversarial samples.
> However, we also acknowledge that the improvement is not substantial, indicating that our method's effectiveness does not heavily rely on hyperparameter selection, showcasing higher generalization and fidelity. We also provide the performance results utilizing different transferable attack methods in the appendix, which proves the effectiveness of our method.
>
> Concerning the $\epsilon$ value, the value of 8 in ablation experiments is not large as we have followed the literature method from [2]. Typically, only when $\epsilon<=16$, we consider the samples can reliably cross more general decision boundaries for a satisfactory attack outcome. We have now conducted more additional experiments to demonstrate that our method doesn’t need particular hyperparameter selection.
>
> For the INFD score and sanity check, our method is faithful to the underlying model, as demonstrated in Section 3 of IG [3], which has proven the Completeness (Sensitivity) axiom as a sanity check. We did not utilize Faithfulness [4] as a fidelity metric because its calculation involves random selection of a subset, and the subset's size and randomness in selection will introduce significant bias to the results. To mitigate this bias, a more extensive range of sampling is required, leading to substantial computational costs.
> Thus, we have been mindful when employing such a stochastic evaluation metric, and we would like to note that for AGI, BIG, GIG and similar methods, they also do not adopt this metric in their experiment.
>
> In the global official comment, we provide INFD score results [5] in Table *2, and conduct ablation experiments with lower $\epsilon$ values in Table *3, demonstrating that our method exhibits the best performance of robustness and fidelity.
>
> Reference:
>
> [1] Zhu, Z., Chen, H., Zhang, J., Wang, X., Jin, Z., Lu, Q., ... & Choo, K. K. R. (2023, October). Improving Adversarial Transferability via Frequency-based Stationary Point Search. In Proceedings of the 32nd ACM International Conference on Information and Knowledge Management (pp. 3626-3635).
>
> [2] Zhang, J., Wu, W., Huang, J. T., Huang, Y., Wang, W., Su, Y., & Lyu, M. R. (2022). Improving adversarial transferability via neuron attribution-based attacks. In Proceedings of the IEEE/CVF Conference on Computer Vision and Pattern Recognition (pp. 14993-15002).
>
> [3] Sundararajan, M., Taly, A., & Yan, Q. (2017, July). Axiomatic attribution for deep networks. In International conference on machine learning (pp. 3319-3328). PMLR.
>
> [4] Bhatt, U., Weller, A., & Moura, J. M. (2021, January). Evaluating and aggregating feature-based model explanations. In Proceedings of the Twenty-Ninth International Conference on International Joint Conferences on Artificial Intelligence (pp. 3016-3022).
>
>  [5] Yeh, C. K., Hsieh, C. Y., Suggala, A., Inouye, D. I., & Ravikumar, P. K. (2019). On the (in) fidelity and sensitivity of explanations. Advances in Neural Information Processing Systems, 32.

---

> > ### Comment · Reviewer_H27T · 2023-11-23
> > **My score has to remain**
> >
> > Thanks for the exhaustive responses provided by the authors. After reading the rebuttal, I have a mixed feeling about this paper.
> >
> > First of all, I acknowledge the authors' efforts in providing many more tables in the global comment. However, after I read the rebuttal and the table more carefully, I do not think they managed to address my concerns.
> >
> > **Motivation.**
> > > The motivation is to enhance the attribution performance by considering different decision boundaries with the insights from transferable adversarial attacks. We consider the features to be crucial for the model output if the altered sample will cross the decision boundary. The summarization of the contributions of these features is considered as the attribution results. ...
> >
> > I am still unable to clearly see "why one wants to cross multiple decision boundary". What benefit does crossing multiple decision boundary bring to our explanation. This is too hand-wavy. I really hope there is any rigorous justification.
> >
> > > In this way, we consider the obtained sample will be able to cross more decision boundaries without the parameters from the target model, which means a more accurate attribution result.
> >
> > I don't think transferable adversarial examples play any role in motivating this paper. Transferable attack is a **means** to find baseline points that on the other sides of many decision boundaries. That is, it is not a motivation. The technique itself does not justify any purpose.
> >
> > **Generalization**
> >
> > Thanks for showing FDE actually transfer to many other models. However, one weird thing to me is that FDE is so much better than your baselines. To be honest, the purpose of this table is not to show FDE works better than other methods (otherwise I have to cite more recent papers and show that you are not using reasonable baselines), which is not the reason I asked for this experiment. The authors can add the results of FDE to the paper but you should not try to demonstrate how well it is compared to your baselines. Namely, the purpose of this experiment is to show you use some method that transfer reasonably good and it is not necessary to be the state-of-the-art attack to be useful to your proposed explanation.
> >
> > **Evaluations**
> >
> > Faithfulness is not equal to completeness. But INFD results show that your proposed method is reasonably faithful within some ball park compared to BIG and Saliency Map. I appreciate for this result.
> >
> > However, on the other hand, Table 3 strengthens my concerns about whether "using FDE points" is actually important to your final results. No matter how you change $\epsilon$, your results do not change. I do not interpret $\epsilon$ as a hyper-parameter. It is a really important argument. It determines how far the point you choose is away from the input of interest. And both results show that that distance does not matter at all, which is counter-intuitive.
> >
> > In summary, my score is mostly based on motivations and take-aways of the current paper. I appreciate for running more experiments during this period. However, I find too many things are not clearly stated, discussed, justified and presented. As a result, I cannot increase my score.

---

> ### Author Response · Authors · 2023-11-23
>
> Thanks to the reviewer for the response.
>
> **Clearer Motivation clarification:**
>
> It is obvious that if an input feature can cross the decision boundary of the model and change the decision-making of the model, then this input feature can be attributed as a high-contribution feature, that is, an important feature. However, we need to consider the impact of inaccurate decision boundaries during model training, because usually, the confidence of the training data is high and far away from the decision boundaries. That is to say, those samples that are very close to the decision boundary are likely to be OOD samples and are prone to overfitting.
>
> We believe that since the transferable attacks can more generally cross the decision boundary of the target model, the decision boundary obtained by the transferable attacks is likely to be less overfitting than the original decision boundary, in other words, more accurate. Therefore, it is clear whether we can use the characteristics of transferable attacks as our motivation for attribution. Because the relationship between attribution and decision boundaries is very obvious. We also illustrate the equivalence between decision boundaries and input exploration in the paper.
>
> **Clearer Generalization clarification:**
>
> Regarding your **original comment**, *'I am pretty worried the authors explain this conclusion as altering some features in the frequency domain helps to create examples that transfer better, especially for methods just proposed in this paper (Eq. 4 - 5). Have you verified your adversarial examples actually transfer better? Can you provide some analytical results convincing me this helps better explore more decision boundaries?'*
>
> Our purpose in adding FDE experiments is to **implement the proposed requirements**, *which also shows that our method can achieve better transferable attack results (meaning more decision boundaries are crossed)*. We do not attempt to prove that the transferability of FDE is better, since our goal is focusing on attribution. The cited baselines are from recent years to illustrate that there is indeed an impetus to crossing more general baselines. Similarly, comparing all transferable attack methods is not realistic. For us, we have selected the **most representative and SOTA works** to prove the effectiveness of FDE. Our goal is not to create a better transferable attack but to illustrate the effectiveness of frequency exploration for attribution. Always, there will be better ways to move the attacks, but that's not the focus of this paper, not either the job for attribution.
>
> For the comparison process, we used the same open-source code and the same experimental design as the baselines in most recent years. And we have added all the details in the anonymous code.
>
> **Clearer clarification of Evaluations:**
>
> Our experiments show that the larger $epsilon$ does have an impact on the results, and it shows an increasing trend. Together with Table 5 (in our submission), Table *5, and the Table *3, the impact of $epsilon$ cannot be ignored. From the literature, the general setting for $epsilon$ is 16, which is actually larger than 8. In addition, our attribution process also requires multiple gradient ascents, so even a very low $epsilon$ will have an effect after multiple gradient ascents. This is not counter-intuitive.

---

### Author Response · Authors · 2023-11-15
**Global comment for experiments (Part 1)**

Table *1-1. The attack success rate of the non-defensive training model when using Inception-v3 as the source model

| Method | Inc-v3 | Inc-v4 | IncRes-v2 | ResNet-50 | ResNet-101 | ResNet-152 |
|--------|--------|--------|-----------|-----------|------------|------------|
| DI-FGSM     | **100.00%** | 27.80% | 20.20% | 24.30% | 21.90% | 18.40% |
| TI-FGSM     | 99.90% | 27.50% | 19.20% | 23.40% | 20.70% | 18.00% |
| MI-FGSM     | **100.00%** | 50.70% | 47.10% | 45.60% | 40.90% | 41.40% |
| DITI-FGSM   | 99.90% | 28.40% | 19.30% | 23.10% | 20.20% | 18.50% |
| TIMI-FGSM   | 99.90% | 51.50% | 46.30% | 46.40% | 42.60% | 41.90% |
| FDE    | 99.50% | **88.50%** | **86.60%** | **81.30%** | **80.60%** | **80.50%** |

In Table *1-1 and *1-2, FDE stands for frequency domain exoloration, and DI-FGSM, TI-FGSM, MI-FGSM, TIMI-FGSM, and TIDI-FGSM are the variant algorithms representing the state-of-the-art adversarial attack methods as competitive models. We present the attack success rate as the analytical results, in which a higher value represents a higher potential to cross other general decision boundaries.

Table *1-2. The attack success rate of the defensive training model when using Inception-v3 as the source model

| Method | Ens3-Inc-v3 | Ens4-Inc-v3 | Ens-IncRes-v2 |
|--------|-------------|-------------|--------------|
| DI-FGSM     | 11.30%      | 11.30%      | 5.00%        |
| TI-FGSM     | 13.20%      | 13.50%      | 7.10%        |
| MI-FGSM     | 22.90%      | 23.10%      | 10.70%       |
| DITI-FGSM   | 13.40%      | 13.00%      | 5.80%        |
| TIMI-FGSM   | 30.20%      | 31.30%      | 20.60%       |
| FDE    | **74.50%**  | **74.70%**  | **59.90%**   |


Table *2. INFD Score (the lower value indicates the better performance)

 | MODEL        | METHOD         | INFD      |
|--------------|----------------|-----------|
| inception_v3 | AGI            | 3.839     |
| inception_v3 | BIG            | 3.928     |
| inception_v3 | DeepLIFT        | 110.158   |
| inception_v3 | EG             | 111.631   |
| inception_v3 | Fast IG        | 111.440   |
| inception_v3 | GIG            | 37.670    |
| inception_v3 | IG             | 66.509    |
| inception_v3 | **AttEXplore**     | **3.728**     |
| inception_v3 | Saliency Map   | 4.078     |
| inception_v3 | SG             | 63.659    |
| resnet50     | AGI            | 1.003     |
| resnet50     | BIG            | 0.708     |
| resnet50     | DeepLIFT        | 18.828    |
| resnet50     | EG             | 143.593   |
| resnet50     | Fast IG        | 135.651   |
| resnet50     | GIG            | 39.659    |
| resnet50     | IG             | 85.834    |
| resnet50     | **AttEXplore**     | **0.671**     |
| resnet50     | Saliency Map   | 0.696     |
| resnet50     | SG             | 42.504    |
| vgg16        | AGI            | 0.880     |
| vgg16        | BIG            | 0.498     |
| vgg16        | DeepLIFT        | 9.746     |
| vgg16        | EG             | 220.376   |
| vgg16        | Fast IG        | 211.104   |
| vgg16        | GIG            | 47.988    |
| vgg16        | IG             | 124.474   |
| vgg16        | **AttEXplore**     | **0.600**     |
| vgg16        | Saliency Map   | 0.499     |
| vgg16        | SG             | 72.912    |

Table *3. Ablation experiment when the perturbation rate is less than 8
| MODEL        | EPSILON | N   | INS     | DEL     |
|--------------|---------|-----|---------|---------|
| inception_v3 | 1       | 60  | 0.459   | 0.031   |
| inception_v3 | 2       | 60  | 0.459   | 0.032   |
| inception_v3 | 3       | 60  | 0.459   | 0.032   |
| inception_v3 | 4       | 60  | 0.461   | 0.031   |
| inception_v3 | 5       | 60  | 0.460   | 0.032   |
| inception_v3 | 6       | 60  | 0.461   | 0.031   |
| inception_v3 | 7       | 60  | 0.460   | 0.032   |
| resnet50     | 1       | 60  | 0.397   | 0.031   |
| resnet50     | 2       | 60  | 0.397   | 0.031   |
| resnet50     | 3       | 60  | 0.398   | 0.031   |
| resnet50     | 4       | 60  | 0.401   | 0.032   |
| resnet50     | 5       | 60  | 0.400   | 0.032   |
| resnet50     | 6       | 60  | 0.403   | 0.032   |
| resnet50     | 7       | 60  | 0.403   | 0.032   |
| vgg16        | 1       | 60  | 0.298   | 0.021   |
| vgg16        | 2       | 60  | 0.299   | 0.022   |
| vgg16        | 3       | 60  | 0.300   | 0.022   |
| vgg16        | 4       | 60  | 0.300   | 0.022   |
| vgg16        | 5       | 60  | 0.300   | 0.022   |
| vgg16        | 6       | 60  | 0.302   | 0.022   |
| vgg16        | 7       | 60  | 0.303   | 0.022   |

---

### Author Response · Authors · 2023-11-15
**Global comment for experiments (Part 2)**

Table *4. Experiments on ViT
| MODEL     | METHOD       | INS     | DEL     |
|-----------|--------------|---------|---------|
| vit_b_16  | Saliency Map  | 0.373   | 0.125   |
| vit_b_16  | BIG           | 0.422   | 0.093   |
| vit_b_16  | GIG           | 0.335   | 0.046   |
| vit_b_16  | DeepLIFT      | 0.296   | 0.063   |
| vit_b_16  | EG            | 0.361   | 0.329   |
| vit_b_16  | Fast IG       | 0.216   | 0.071   |
| vit_b_16  | SG            | 0.428   | 0.035   |
| vit_b_16  | AGI           | 0.425   | 0.069   |
| vit_b_16  | AttEXplore    | **0.470**   | **0.062**   |
| vit_b_16  | IG            | 0.346   | 0.051   |
| vit_b_16  | EG            | 0.361   | 0.329   |



Table *5. Ablation experiment when the perturbation rate is 48
| MODEL        | EPSILON | N   | INS     | DEL     |
|--------------|---------|-----|---------|---------|
| inception_v3 | 48      | 1   | 0.459   | 0.028   |
| inception_v3 | 48      | 2   | 0.461   | 0.028   |
| inception_v3 | 48      | 3   | 0.465   | 0.028   |
| inception_v3 | 48      | 4   | 0.465   | 0.027   |
| inception_v3 | 48      | 5   | 0.464   | 0.028   |
| inception_v3 | 48      | 6   | 0.467   | 0.028   |
| inception_v3 | 48      | 7   | 0.468   | 0.027   |
| inception_v3 | 48      | 8   | 0.470   | 0.028   |
| inception_v3 | 48      | 9   | 0.472   | 0.029   |
| inception_v3 | 48      | 10  | 0.466   | 0.028   |
| inception_v3 | 48      | 20  | 0.471   | 0.029   |
| inception_v3 | 48      | 30  | 0.473   | 0.029   |
| inception_v3 | 48      | 40  | 0.473   | 0.030   |
| inception_v3 | 48      | 50  | 0.474   | 0.029   |
| inception_v3 | 48      | 60  | 0.474   | 0.030   |
| resnet50     | 48      | 1   | 0.406   | 0.027   |
| resnet50     | 48      | 2   | 0.414   | 0.028   |
| resnet50     | 48      | 3   | 0.417   | 0.028   |
| resnet50     | 48      | 4   | 0.417   | 0.029   |
| resnet50     | 48      | 5   | 0.419   | 0.030   |
| resnet50     | 48      | 6   | 0.420   | 0.029   |
| resnet50     | 48      | 7   | 0.422   | 0.030   |
| resnet50     | 48      | 8   | 0.422   | 0.030   |
| resnet50     | 48      | 9   | 0.422   | 0.030   |
| resnet50     | 48      | 10  | 0.423   | 0.030   |
| resnet50     | 48      | 20  | 0.425   | 0.031   |
| resnet50     | 48      | 30  | 0.427   | 0.031   |
| resnet50     | 48      | 40  | 0.427   | 0.032   |
| resnet50     | 48      | 50  | 0.428   | 0.032   |
| resnet50     | 48      | 60  | 0.428   | 0.032   |
| vgg16        | 48      | 1   | 0.298   | 0.019   |
| vgg16        | 48      | 2   | 0.304   | 0.019   |
| vgg16        | 48      | 3   | 0.306   | 0.019   |
| vgg16        | 48      | 4   | 0.308   | 0.020   |
| vgg16        | 48      | 5   | 0.308   | 0.020   |
| vgg16        | 48      | 6   | 0.308   | 0.020   |
| vgg16        | 48      | 7   | 0.310   | 0.020   |
| vgg16        | 48      | 8   | 0.310   | 0.020   |
| vgg16        | 48      | 9   | 0.311   | 0.021   |
| vgg16        | 48      | 10  | 0.311   | 0.021   |
| vgg16        | 48      | 20  | 0.312   | 0.021   |
| vgg16        | 48      | 30  | 0.313   | 0.021   |
| vgg16        | 48      | 40  | 0.313   | 0.022   |
| vgg16        | 48      | 50  | 0.313   | 0.022   |
| vgg16        | 48      | 60  | 0.315   | 0.022   |

---

### Meta-Review · Area_Chair_mB6j · 2023-12-15

**Metareview:**

This paper introduces an attribution method for neural network interpretability and illuminates the method relationship to adversarial examples. The main strength is to connect attribution methodology with adversarial examples, which in and of itself is valuable.

However, there are various remaining concerns by reviewers that have not been addressed, but need to be considered. A prominent concern is the motivation of the proposed approach. It is not sufficiently clear what this new method accomplishes other than attempting to re-introduce concepts from the transferable adversarial examples literature in the attribution context. On the other hand reviewers did acknowledge the convincing benchmark results, computational efficiency of the method and the connection to adversarial examples as worthwhile contributions.

Hence, we recommend the paper to be accepted to the conference as a poster presentation.

**Justification For Why Not Higher Score:**

The motivation of the paper is lacking and this makes it difficult to assess the significance and utility of the proposed method. On the other hand, multiple reviewers acknowledged the convincing empirical results and the connections to the adversarial example literature.

**Justification For Why Not Lower Score:**

The reviewers largely agreed that the merits outweigh the flaws in this work

---

### Decision · Program_Chairs · 2024-01-16

Accept (poster)